# SiNGER: A Clearer Voice Distills Vision Transformers Further

**Geunhyeok Yu**[1,*]    **Sunjae Jeong**[1,2,*]    **Yoonyoung Choi**[1]
**Jaeseung Kim**[2]    **Hyoseok Hwang**[1,†]
[1] Kyung Hee University    [2] MOBILTECH CO., LTD
{geunhyeok, choiyy0313, sunj, hyoseok}@khu.ac.kr
jason.kim@mobiltech.io

## ABSTRACT

Vision Transformers are widely adopted as the backbone of vision foundation models, but they are known to produce high-norm artifacts that degrade representation quality. When knowledge distillation transfers these features to students, high-norm artifacts dominate the objective, so students overfit to artifacts and underweight informative signals, diminishing the gains from larger models. Prior work attempted to remove artifacts but encountered an inherent trade-off between artifact suppression and preserving informative signals from teachers. To address this, we introduce **Si**ngular **N**ullspace-**G**uided **E**nergy **R**eallocation (SiNGER), a novel distillation framework that suppresses artifacts while preserving informative signals. The key idea is principled teacher feature refinement: during refinement, we leverage the nullspace-guided perturbation to preserve information while suppressing artifacts. Then, the refined teacher's features are distilled to a student. We implement this perturbation efficiently with a LoRA-based adapter that requires minimal structural modification. Extensive experiments show that SiNGER consistently improves student models, achieving state-of-the-art performance in multiple downstream tasks and producing clearer and more interpretable representations. Code is available at https://github.com/AIRLABkhu/SiNGER.

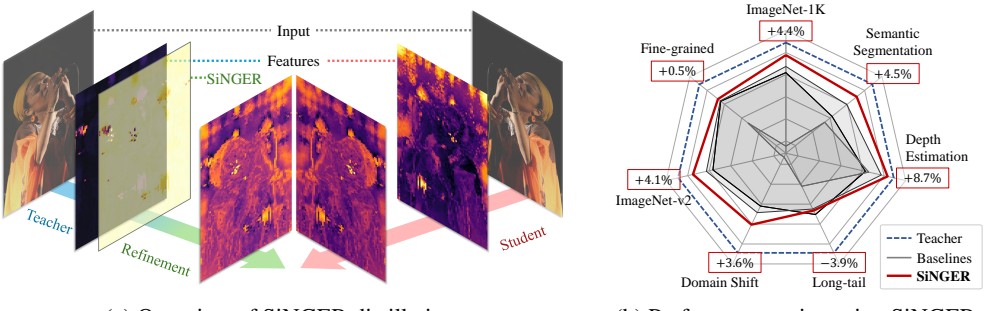

(a) Overview of SiNGER distillation.    (b) Performance gains using SiNGER.

Figure 1: SiNGER suppresses artifacts and enhances transfer. (a) Feature visualizations highlight clearer and more interpretable representations. (b) Radar chart shows consistent multi-task gains.

## 1 INTRODUCTION

Transformers have become the de facto standard architecture in both research and industry due to their scalability and effectiveness (Oquab et al., 2024; Radford et al., 2021). Their token-based self-attention mechanism is broadly applicable with minimal inductive bias (Lu et al., 2022), and has enabled significant advances in computer vision and machine learning (Kim et al., 2024; Sariyildiz et al., 2024). Vision Transformers (ViTs, Dosovitskiy et al. (2021)) extend this paradigm to visual data and form the backbone of Vision Foundation Models (VFMs). Compared to convolutional networks, ViTs rely less on spatial inductive biases and instead exploit scale to achieve high

---

[*]Equally contributed.
[†]Corresponding author.

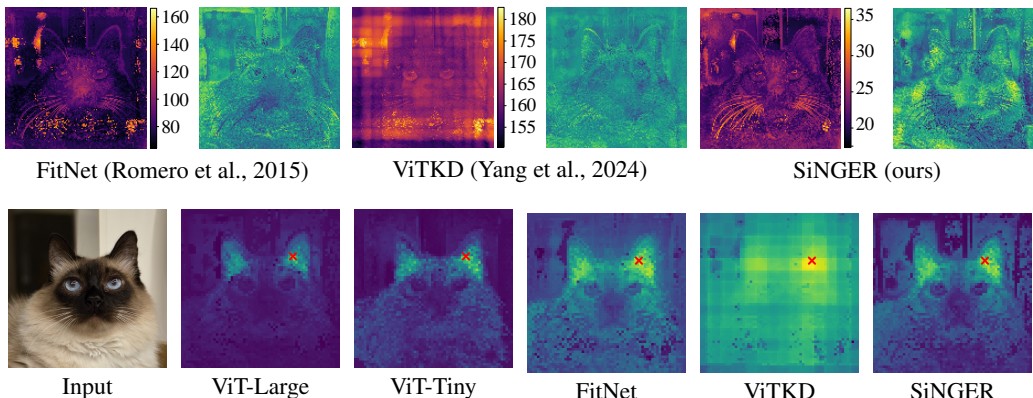

Figure 2: Qualitative analysis. **Row 1**: KD method comparison. Left: distilled feature map colored by patch norm, Right: patch-wise cosine similarity to the teacher. **Row 2**: Input image, two pretrained ViTs, and three ViT-L → ViT-T distilled variants. Each panel shows similarity from the ×-marked patch. SiNGER most closely preserves teacher semantics, showing the most coherent teacher-consistent similarity patterns.

performance. ViTs are also highly scalable since supervised or self-supervised training produces increasingly generalizable representations (Oquab et al., 2024; Touvron et al., 2022). However, the quadratic complexity of self-attention severely limits the practicality of scaling ViTs. This tension between accuracy and efficiency motivates the study of compression. Pruning (Yang et al., 2023) and quantization (Liu et al., 2021) have been explored, but pruning often fails to deliver practical speedup due to structural rigidity (Aghli & Ribeiro, 2021), and quantization can induce numerical instability (Jiang et al., 2024; Javed et al., 2024). Knowledge distillation (KD, Hinton et al. (2015)) has emerged as the most reliable solution for transferring knowledge from large ViTs to smaller students. KD methods span diverse targets and frameworks (Sun et al., 2024; Romero et al., 2015; Ranzinger et al., 2024), consistently yielding structurally and numerically stable compact models.

Nevertheless, KD for ViTs suffers from subtle but critical limitations in their representation space. Darcet et al. (2024) revealed that ViT token representations contain high-norm artifacts. Wang et al. (2025) argue that these artifacts are singular defects induced by power-iteration-like accumulation across residual blocks, whereby tokens align with the leading left singular vector of the pre-trained weights. These artifacts interact poorly with the standard feature mean squared error objective in KD: when the teacher and student are matched, gradients concentrate on the few high-norm tokens, producing an outlier-driven optimization bias that obscures informative signals in the inlier structure. Therefore, suppressing outlier norms in teacher features is essential for KD in ViTs as the scale grows. Prior work mitigated this issue via random masking of teacher features (Yang et al., 2024); however, this inevitably removes informative signals. Therefore, a key challenge is to mitigate these artifacts without losing valuable information, a fundamental trade-off that requires a principled approach.

To resolve this trade-off, we introduce a nullspace-guided suppression: we modify only the nullspace component in the teacher features, mathematically, the subspace orthogonal to the downstream space. This yields student–optimal supervision by suppressing artifacts without sacrificing informative signals. Based on this insight, we propose **Si**ngular **N**ullspace-**G**uided **E**nergy **R**eallocation (SiNGER), a framework that addresses this trade-off in ViTs distillation, illustrated in Figure 1a. To minimize the modification to the teacher's signal, we attach a lightweight LoRA-based adapter (Hu et al., 2022) to the KD architecture, which refines the teacher features. The adapter produces a minimal perturbation guided toward the left-nullspace of the next block, suppressing high-norm outliers while leaving the next block output unchanged. Our method achieves superior performance compared to baselines across multiple downstream tasks (Figure 1b). It also produces more structured and interpretable feature maps (Figure 2). Our contributions are summarized as follows:

- We propose a novel distillation framework (SiNGER) that refines teacher signals via the LoRA-based adapter with nullspace initialization to guide effective perturbations.

- We analyze a fundamental limitation of naïve ViT distillation, showing degraded transfer on downstream benchmarks along with qualitative evidence.
- We provide extensive ablation studies to analyze the contribution of each component in SiNGER and validate the robustness of our framework.
- We demonstrate through extensive experiments that our method exceeds baseline performance across tasks and produces more interpretable feature maps.

## 2 RELATED WORKS

**Vision Transformers.** Visual transformers (Dosovitskiy et al., 2021; Touvron et al., 2022) underpin many VFMs and have become the representative architecture for large-scale visual learning. Unlike convolutional networks (He et al., 2016; Howard et al., 2017), ViTs rely on self-attention with fully connected layers. Since ViTs form the architectural core of most VFMs, studying them provides representative insights that generalize broadly. Similarly, parameter-efficient tuning methods such as LoRA (Hu et al., 2022) highlight how minimal perturbations can effectively adapt VFMs, a perspective that motivates our artifact-suppressing perturbations. However, the quadratic complexity of the ViT architecture limits the practicality of large models despite their advanced representation.

**Knowledge Distillation.** KD compresses models by training a smaller student to mimic a larger teacher (Hinton et al., 2015). Among various approaches, FitNet-style methods (Romero et al., 2015) that align intermediate features are especially influential, as they encourage the student to learn useful representations beyond logits (Sun et al., 2024). Later extensions incorporated relational structures (Park et al., 2019) or multi-teacher settings (Ranzinger et al., 2024), while adaptations for ViTs (Touvron et al., 2022) aimed to respect their architectural characteristics. Despite these advances, distillation applied to VFMs often inherits undesirable properties from teachers, revealing the need for methods that improve not only compression but also the quality of transferred representations.

**Artifacts in Transformers.** Artifacts are a recurring issue in visual transformer models, degrading the representation quality. Darcet et al. (2024) demonstrated that ViTs produce high-norm artifacts, particularly in background regions, harming interpretability and dense prediction. Practical suppression strategies include register tokens (Darcet et al., 2024), and recent work argues these artifacts arise from power-method-like accumulation across residual layers, aligning tokens with the leading left singular vector (Wang et al., 2025). In the knowledge distillation domain, ViTKD (Yang et al., 2024) randomly masks teacher features to reduce the mimicking of high-norm artifacts. However, such indiscriminate masking also removes informative inlier signals, motivating artifact-aware KD that suppresses high-norm artifacts while preserving inlier structure.

## 3 METHOD

### 3.1 PROBLEM FORMULATION

#### 3.1.1 HIGH-NORM OUTLIERS IN VISION TRANSFORMERS

In large ViTs, a non-negligible fraction of patch features in $F_l^{\mathrm{T}}$ exhibit high-norm artifacts (outliers). Their prevalence and magnitude increase with model capacity, as Darcet et al. (2024) reported. This is particularly consequential for distillation from a larger teacher to a smaller student: artifact-prone teacher features introduce a systematic imbalance at the feature level and can obscure informative signals.

#### 3.1.2 KNOWLEDGE DISTILLATION OBJECTIVE

Let $F_l^{\mathrm{T}} \in \mathbb{R}^{n \times d^{\mathrm{T}}}$ and $F_l^{\mathrm{S}} \in \mathbb{R}^{n \times d^{\mathrm{S}}}$ denote teacher (T) and student (S) features at layer $l$. We align dimensions $d^{\mathrm{S}} \to d^{\mathrm{T}}$ with a trainable projection $P_l : \mathbb{R}^{d^{\mathrm{S}}} \to \mathbb{R}^{d^{\mathrm{T}}}$ and define the feature-level KD loss as follow:

$$\mathcal{L}_{\mathrm{KD},l} = \frac{1}{n} \sum_{i=1}^{n} \|F_{l,i}^{\mathrm{T}} - P_l(F_{l,i}^{\mathrm{S}})\|^2, \tag{1}$$

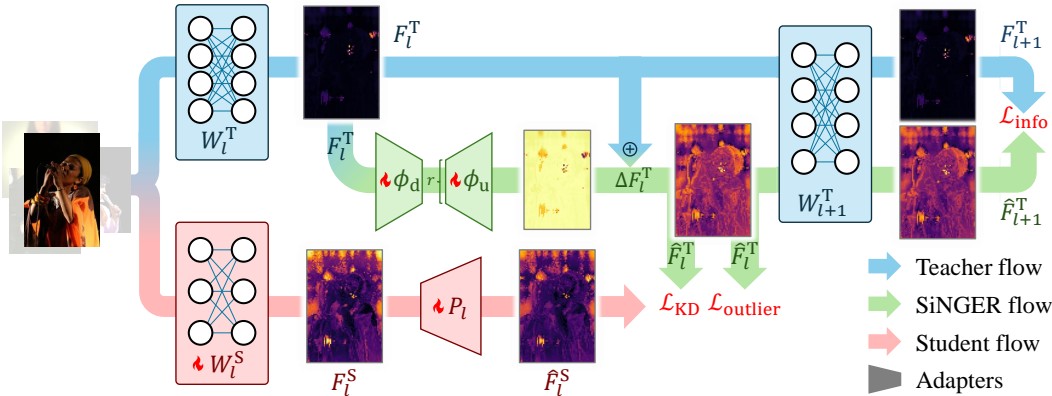

Figure 3: The overall pipeline of knowledge distillation with the SiNGER adapter at $l$th layer .

where $F_{l,i}^{\mathrm{T}}$ and $F_{l,i}^{\mathrm{S}}$ denote the $i$-th patch feature and $n$ is the number of patches and $\|\cdot\|$ means the $\ell_2$-norm.

### 3.1.3 OUTLIER DOMINANCE AND GRADIENT BIAS

Partition the patch indices into an outlier set $\mathcal{O}_l$ and an inlier set $\mathcal{I}_l$ to obtain

$$\mathcal{L}_{\mathrm{KD},l} = \frac{1}{n}\left(\underbrace{\sum_{i\in\mathcal{O}_l}\|F_{l,i}^{\mathrm{T}} - P_l(F_{l,i}^{\mathrm{S}})\|^2}_{\text{Outlier Term}} + \underbrace{\sum_{j\in\mathcal{I}_l}\|F_{l,j}^{\mathrm{T}} - P_l(F_{l,j}^{\mathrm{S}})\|^2}_{\text{Inlier Term}}\right). \tag{2}$$

By construction, for $i \in \mathcal{O}_l$ we have $\|F_{l,i}^{\mathrm{T}}\| \gg \|F_{l,j}^{\mathrm{T}}\|$ for $j \in \mathcal{I}_l$. Hence, when the residual magnitudes are of similar order across patches, the outlier term dominates both the objective and its gradients. In particular,

$$\nabla_{P_l(F_{l,i}^{\mathrm{S}})}\mathcal{L}_{\mathrm{KD},l} = \frac{2}{n}\big(P_l(F_{l,i}^{\mathrm{S}}) - F_{l,i}^{\mathrm{T}}\big), \tag{3}$$

so outliers induce proportionally larger updates. Optimization is therefore biased toward mimicking a few high-norm outliers, rather than consolidating the majority inlier structure that carries most of the informative signals. This gradient bias disrupts the learning of the dominant inlier representation and leads to suboptimal transfer. We therefore seek to refine the teacher features $F_l^{\mathrm{T}}$ at layer $l$ before distillation, so that they are more conducive to transferring informative signals to the student.

### 3.2 SINGULAR NULLSPACE-GUIDED ENERGY REALLOCATION

We consider KD at layer $l$, where high-norm outliers in $F_l^{\mathrm{T}}$ induce gradient bias toward a few tokens. To prevent this, the outlier term in Equation (2) must be weakened, which in practice means reducing the norm of outlier patches in $F_l^{\mathrm{T}}$. However, naïve shrinkage erodes information carried by the larger teacher and can nullify the benefits of distillation.

#### 3.2.1 PERTURBATION ON NULLSPACE

Let $F_l^{\mathrm{T}} \in \mathbb{R}^{n\times d^{\mathrm{T}}}$ and define a refined feature map $\hat{F}_l^{\mathrm{T}} = F_l^{\mathrm{T}} + \Delta F_l^{\mathrm{T}}$. Our two objectives are:

1. **Suppress Outlier Norms.** Reduce the norm of high-norm patches in $F_l^{\mathrm{T}}$ (Figure 4a).
2. **Preserve Information.** Ensure that when the modified features are fed into the next teacher block, the conveyed information is not altered (Figure 4b).

Consider the next block at layer $l+1$ with transformation $W_{l+1} \in \mathbb{R}^{d^{\mathrm{T}}\times d^{\mathrm{T}}}$. Then $\hat{F}_l^{\mathrm{T}}$ preserves the next-block output if and only if

$$\big(F_l^{\mathrm{T}} + \Delta F_l^{\mathrm{T}}\big)W_{l+1} = \hat{F}_l^{\mathrm{T}}W_{l+1} \iff \Delta F_l^{\mathrm{T}}W_{l+1} = \mathbf{0}. \tag{4}$$

A perturbation $\Delta F_l^{\mathrm{T}}$ that satisfies the above is obtained by restricting it to the left-nullspace of the next block $W_{l+1}$. Let us $N_{l+1} := \mathrm{Null}\big((W_{l+1})^{\top}\big)$ denote this left-nullspace. Then the requirement is as follows:

$$\mathrm{row}\big(\Delta F_l^{\mathrm{T}}\big) \subseteq N_{l+1}. \tag{5}$$

Consequently, to allow effective distillation, we refine the features of the teacher $F_l^{\mathrm{T}}$ to $\hat{F}_l^{\mathrm{T}}$ by a perturbation guided to the left-nullspace $N_{l+1}$.

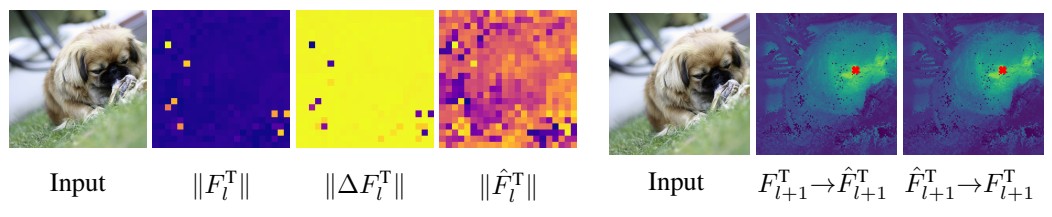

| Input | $\|F_l^{\mathrm{T}}\|$ | $\|\Delta F_l^{\mathrm{T}}\|$ | $\|\hat{F}_l^{\mathrm{T}}\|$ | Input | $F_{l+1}^{\mathrm{T}} \rightarrow \hat{F}_{l+1}^{\mathrm{T}}$ | $\hat{F}_{l+1}^{\mathrm{T}} \rightarrow F_{l+1}^{\mathrm{T}}$ |

(a) Outlier suppression with the proposed adapter.  (b) Information preservation after the $l+1$th layer.

Figure 4: Two objectives of SiNGER; (a) outlier suppression and (b) information preservation. $\|\Delta F_l^{\mathrm{T}}\|$ in (a) is signed with the cosine-similarity between $\Delta F_l^{\mathrm{T}}$ and $F_l^{\mathrm{T}}$. In (b), the cosine similarity between $\times$-marked patch and every patch of another feature map is visualized.

### 3.2.2 ADAPTER-BASED FEATURE REFINEMENT

We refine $F_l^{\mathrm{T}}$ by adding a low-rank perturbation produced by a LoRA-based adapter while freezing all teacher weights.

$$\hat{F}_l^{\mathrm{T}} = F_l^{\mathrm{T}} + \Delta F_l^{\mathrm{T}}, \qquad \Delta F_l^{\mathrm{T}} = \big(F_l^{\mathrm{T}} \phi_{\mathrm{down},l}\big) \phi_{\mathrm{up},l}, \tag{6}$$

where $\phi_{\mathrm{down},l} \in \mathbb{R}^{d^{\mathrm{T}} \times r}$, $\phi_{\mathrm{up},l} \in \mathbb{R}^{r \times d^{\mathrm{T}}}$, and $r \ll d^{\mathrm{T}}$.

To bias $\Delta F_l^{\mathrm{T}}$ toward the left-nullspace $N_{l+1}$ of $W_{l+1}$, we set the initial weights of adapter,

$$\phi_{\mathrm{down},l} := N_{l+1}, \qquad \phi_{\mathrm{up},l} := N_{l+1}^{\top}. \tag{7}$$

This initialization guides the optimization to remain near $N_{l+1}$ and to find solutions that satisfy the two objectives. Because the next block is nonlinear, its exact nullspace cannot be obtained via SVD. We adopt a practical linearization of the next block, $W_{l+1} \approx \tilde{W}_{l+1}$, and define $\tilde{N}_{l+1}$ as the left singular vectors associated with the $r$ smallest singular values of $\tilde{W}_{l+1}$. By construction, $\tilde{N}_{l+1}$ collects the left singular vectors corresponding to the $r$ smallest singular values of $\tilde{W}_{l+1}$, so $\|\tilde{N}_{l+1}^{\top} \tilde{W}_{l+1}\| = \sigma_{d-r+1}$. Moreover, Appendix A provides a detailed spectral analysis (sublayer perturbations, singular value diagnostics, and $\varepsilon$-null bounds) showing that the same approximate-null relation holds for the nonlinear block. Consequently,

$$\phi_{\mathrm{down},l} := \tilde{N}_{l+1}, \qquad \phi_{\mathrm{up},l} := \tilde{N}_{l+1}^{\top}. \tag{8}$$

### 3.3 KNOWLEDGE DISTILLATION WITH SiNGER

Figure 3 summarizes the pipeline: SiNGER refines teacher features at selected layers before feature matching. Let $\mathcal{D} = l_{\mathrm{inter}} \cup \{l_{\mathrm{final}}\}$ denote the distillation layers, where $l_{\mathrm{inter}}$ is a set of intermediate layers and $l_{\mathrm{final}}$ is the final layer. For each $l \in \mathcal{D}$, an adapter $\phi_l$ transforms $F_l^{\mathrm{T}}$ into $\hat{F}_l^{\mathrm{T}}$. Training is guided by three losses, aggregated over $\mathcal{D}$.

**Knowledge-Distillation Loss.** The student is trained to mimic the refined teacher with $\mathcal{L}_{\mathrm{KD}}$.

$$\mathcal{L}_{\mathrm{KD}} = \sum_{l \in \mathcal{D}} \mathrm{MSE}\big(\hat{F}_l^{\mathrm{T}}, \, P_l(F_l^{\mathrm{S}})\big). \tag{9}$$

**Outlier Suppression Loss.** Adapters are explicitly encouraged to suppress high-norm artifacts. For each $l$, let $\mathcal{O}_l$ be the indices of patches in $\hat{F}_l^{\mathrm{T}}$ whose norms exceed the $\alpha$-percentile $q_{\alpha,l}$ as Equation 10.

$$\mathcal{L}_{\mathrm{outlier}} = \sum_{l \in \mathcal{D}} \frac{1}{|\mathcal{O}_l|} \sum_{i \in \mathcal{O}_l} \big( \|\hat{F}_{l,i}^{\mathrm{T}}\|_2 - q_{\alpha,l} \big)^2 \tag{10}$$

**Information Preservation Loss.** To retain informative signals while suppressing norms, we align feature directions via Gram matching. Define

$$\mathcal{L}_{\text{info},l} = \begin{cases} \text{MSE}\left(G(\hat{F}_{l+1}^{\text{T}}),\ G(F_{l+1}^{\text{T}})\right), & l \in l_{\text{inter}}, \\ \text{MSE}\left(G(\hat{F}_{l}^{\text{T}}),\ G(F_{l}^{\text{T}})\right), & l = l_{\text{final}}. \end{cases} \tag{11}$$

where $G(F)$ denotes the Gram matrix of $F$. This preserves the directional structure passed to the next block for intermediate layers, and preserves the final-layer structure at $l_{\text{final}}$. Consequently, the total information term is $\mathcal{L}_{\text{info}} = \sum_{l \in \mathcal{D}} \mathcal{L}_{\text{info},l}$.

**Training Objective.** We jointly optimize the student parameters $\boldsymbol{\theta}_{\text{S}}$, projection parameters $\boldsymbol{\theta}_P = \{P_l\}_{l \in \mathcal{D}}$, and SiNGER adapter parameters $\boldsymbol{\theta}_\phi = \{\phi_{\text{down},l}, \phi_{\text{up},l}\}_{l \in \mathcal{D}}$ with a single weighted sum loss:

$$\mathcal{L}_{\text{total}} = \mathcal{L}_{\text{KD}} + \lambda_{\text{outlier}}\mathcal{L}_{\text{outlier}} + \lambda_{\text{info}}\mathcal{L}_{\text{info}}, \tag{12}$$

where $\lambda_{\text{outlier}}$ and $\lambda_{\text{info}}$ balance artifact suppression and information alignment. This objective encourages effective transfer while controlling high-norm artifacts in teacher features.

## 4 EXPERIMENTS AND ANALYSIS

### 4.1 DETAILS

**Downstream Tasks.** To evaluate SiNGER-distilled ViT as a VFM, we adopt the student network to a diverse set of downstream tasks. Specifically, we consider six representative benchmarks: ImageNet-1K validation set for large-scale classification (Deng et al., 2009), ADE-20K for semantic segmentation (Zhou et al., 2019), NYUd-v2 for depth estimation (Ignatov et al., 2024), iNaturalist-2019 for long-tail classification (Van Horn et al., 2018), ImageNet-R and ImageNet-v2 for domain shift robustness (Hendrycks et al., 2021; Recht et al., 2019), and four fine-grained classification datasets (Maji et al., 2013; Parkhi et al., 2012; Bossard et al., 2014; Nilsback & Zisserman, 2006).

**Distillation Setup.** We evaluate SiNGER on multiple teacher–student configurations spanning both the canonical ViT (Dosovitskiy et al., 2021) and the modern DeiT-III (Touvron et al., 2022), covering a range of model scales. The detailed architectural specifications are summarized in Appendix B. Student layers are aligned with every second teacher layer. The official implementation of FitNet and ViTKD employs task-specific loss objectives, such as cross-entropy minimization for classification. In contrast, we target VFM distillation and therefore exclude task-specific losses, distilling the last hidden layer's representation.

**Rationale.** We aim to probe pre-training agnostic mechanisms of artifact formation and suppression. To this end, we conduct the full ablation suite on canonical ViTs, whose transparent design and widely adopted training recipe allow tighter control, clearer causal attribution, and more reproducible analysis.

### 4.2 MULTI-TASK EVALUATION

Table 1 summarizes multi-task linear evaluation results across ten benchmarks. The teacher (Large) achieves strong performance, while the Tiny baseline shows significant degradation, particularly on dense prediction tasks; ADE-20K and NYUd-v2. FitNet improves over the Tiny baseline by transferring intermediate features, but still inherits artifacts from the teacher, limiting overall gains. ViTKD performs poorly across all tasks in most cases, as its random masking strategy often collapses feature representations and prevents effective learning. Distillation across diverse teacher-student pairs is discussed in Appendix C and additional discussion on ViTKD is detailed in Appendix D.

By contrast, SiNGER demonstrates consistent improvements over FitNet and ViTKD on most benchmarks. On IN-val, ADE-20K, NYUd-v2, DS, and FG, SiNGER yields large gains, approaching teacher performance despite the smaller capacity. The only exception is iNat2019, where performance slightly drops compared to the non-distilled student-size models, which we attribute to the long-tail nature of the dataset, as Zhang et al. (2023) pointed out. We report an entropy-driven analysis on iNat2019 in Appendix J. Overall, these results confirm that suppressing artifacts during distillation produces student models that are both more accurate and more generalizable across diverse tasks.

| Model | Distillation | IN-val top-1 (↑) | ADE-20K mIoU (↑) | NYUd-v2 RMSE (↓) | iNat2019 top-1 (↑) | DS top-1 (↑) | FG top-1 (↑) |
|---|---|---|---|---|---|---|---|
| ViT-L | | 79.58 | 26.57 | 0.9157 | 71.42 | 54.20 | 82.27 |
| ViT-S | (Non-distilled) | 76.05 | 19.57 | 1.1065 | 65.28 | 44.48 | 77.39 |
| ViT-T | | 58.03 | 14.20 | 1.1807 | 43.95 | 28.75 | 62.02 |
| ViT-L → ViT-S | FitNet | 72.50 | 25.15 | 0.9903 | 52.12 | 40.21 | 71.13 |
| | SiNGER | **79.13** | **30.06** | **0.9026** | **57.21** | **48.91** | **77.59** |
| | Δ | +6.63 | +4.91 | +0.0877 | +5.09 | +8.70 | +6.46 |
| ViT-L → ViT-T | FitNet | 62.43 | 18.73 | 1.0093 | 40.02 | 32.32 | 62.48 |
| | ViTKD | 5.07 | 11.92 | 1.1903 | 23.69 | 2.08 | 33.52 |
| | SiNGER | **70.59** | **21.76** | **0.9406** | **41.11** | **38.87** | **64.61** |
| | Δ | +8.16 | +3.03 | +0.0687 | +1.09 | +6.55 | +2.13 |
| DeiT-III-L | | 84.10 | 26.11 | 1.2311 | 57.59 | 56.95 | 75.35 |
| DeiT-III-B | (Non-distilled) | 82.50 | 25.14 | 1.1781 | 56.41 | 54.73 | 74.65 |
| DeiT-III-S | | 78.90 | 20.28 | 1.2961 | 45.39 | 49.50 | 67.84 |
| DeiT-III-L → DeiT-III-B | FitNet | 60.00 | 26.79 | 1.1625 | 50.04 | 31.53 | 74.78 |
| | ViTKD | 66.54 | 19.58 | 1.2525 | 30.78 | 35.77 | 58.36 |
| | SiNGER | **79.37** | **29.47** | **1.1514** | **53.50** | **49.14** | **75.41** |
| | Δ | +12.83 | +2.68 | +0.0111 | +3.46 | +13.37 | +0.63 |
| DeiT-III-S → DeiT-III-T | FitNet | 51.41 | 16.06 | 1.2920 | 32.81 | 23.74 | 58.53 |
| | ViTKD | 35.56 | 8.71 | 1.4487 | 19.93 | 15.31 | 45.13 |
| | SiNGER | **63.50** | **20.67** | **0.9827** | **37.68** | **32.28** | **64.32** |
| | Δ | +12.09 | +4.61 | +0.3093 | +4.87 | +8.54 | +5.79 |

Table 1: Multi-task linear evaluation results. ImageNet-1K validation (IN-val) for large-scale classification, ADE-20K for semantic segmentation, NYUd-v2 for monocular depth estimation, iNaturalist2019 (iNat2019) for long-tail learning, ImageNet-R and ImageNet-v2 for domain shift (DS), and four fine-grained classification (FG) benchmarks: FGVC-Aircraft, Oxford-IIIT Pet, Food-101, and Flowers-102 were tested. Δ rows indicate the performance gains of SiNGER, computed against the best-performing baseline among the distilled students (underlined).

## 4.3 REPRESENTATION QUALITY

We assess the quality and interpretability of distilled representations by comparing the feature maps and their Gram matrices. Figure 5 depicts the Gram matrices of the feature maps. Quantitatively, SiNGER's Gram matrix is the most similar one to the teacher's Gram matrix. Gram Distance (GD), defined as $\ell_2$ distance between the Gram matrices, confirms this trend Figure 2. This shows that when artifacts are distilled, it disrupts the transfer of patch-wise relation, resulting in degraded student representation. Centered Kernel Analysis (CKA, Kornblith et al. (2019)) measures the linear correlation between two feature maps. FitNet and ViTKD achieve higher similarity by following the teacher too closely, but this reflects replication of artifacts rather than useful knowledge transfer. By contrast, SiNGER learns structurally consistent yet information-preserved representations, balancing similarity with the teacher.

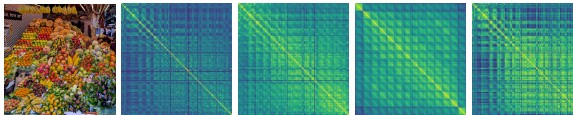

| Metric | FitNet | ViTKD | SiNGER |
|---|---|---|---|
| GD | 0.237 | 0.520 | **0.130** |
| CKA | 0.732 | 0.745 | **0.660** |

Figure 5: Gram matrices of the patches. The input, the teacher, and distilled features; FitNet, ViTKD, and SiNGER in order.

Table 2: The teacher-student representation's similarity in terms of $\ell_2$ distance between the Gram Distance (GD) and CKA.

## 4.4 ADAPTER OPERATION

We empirically analyze how the optimized adapter operates on ImageNet-1K. To probe the coupling with the next layer, we evaluate at an intermediate layer $l = 17$.

**Patch-Norm Distribution Between $F_l$ and $\hat{F}_l$.** We visualize the distribution of patch $l_2$ norms for $F_l$ and $\hat{F}_l$ with side-by-side box plots (Figure 6). The teacher produces high-norm artifacts that are distinctly gathered as a group. We observed that SiNGER effectively draws such artifacts into the

normal-patch range while preserving informative features. This results in stabilized gradient flow through the normal patches.

**Cosine Similarity Between $F_{l+1}$ and $\hat{F}_{l+1}$.** To assess information preservation, we compute patch-wise cosine similarities for both $F_l$ vs. $\hat{F}_l$ and $F_{l+1}$ vs. $\hat{F}_{l+1}$, aggregating per image across the dataset (see Table 3). The $17, 18$-th layers yield cosine similarity of $0.9566$ and $0.9731$ with negligible variance, respectively, which are clearly very similar.

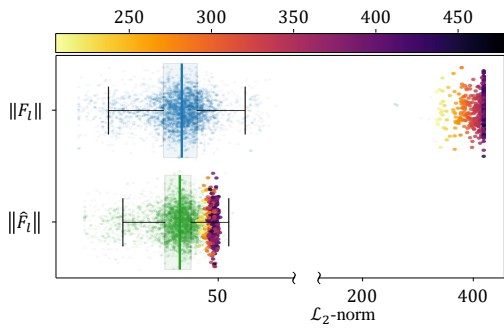

Figure 6: Patch-norm distributions of $F_l$ and $\hat{F}_l$. Artifacts are scale-colored separately with `inferno`.

Table 3: Patch-wise cosine similarity (per-image mean over patches) on ImageNet-1K at $l=17$ and $l+1=18$.

| Pair | $\mu$ | $\sigma$ | median |
|------|-------|----------|--------|
| $F_{17}$ vs. $\hat{F}_{17}$ | 0.9566 | 0.0038 | 0.9569 |
| $F_{18}$ vs. $\hat{F}_{18}$ | 0.9731 | 0.0021 | 0.9732 |

## 4.5 ABLATION STUDY

We report four ablations, focusing on initialization, losses, hyperparameters, and distillation layers. Note that all experiments reported in Table 4 were conducted using a subset of ImageNet-1K.

**Initialization Method.** We validate whether nullspace initialization truly guides the adapter to induce perturbations along the nullspace during optimization by comparing nullspace-biased (SiNGER) and random initializations.

Let the next block at layer $l+1$ be linearized to $\tilde{W}_{l+1} \in \mathbb{R}^{D \times D}$. From $\tilde{W}_{l+1}$, define two rank-$r$ bases: the principal basis $P_{l+1} \in \mathbb{R}^{D \times r}$ (largest singular directions) and the null basis $N_{l+1} \in \mathbb{R}^{D \times r}$ (smallest singular directions). We quantify alignment with the normalized Frobenius norm

| Initialization | $\mathcal{L}_{\text{outlier}}$ | $\mathcal{L}_{\text{info}}$ | IN-val top-1 (↑) | ADE-20K mIoU (↑) | NYUd-v2 RMSE (↓) |
|----------------|--------------------------------|------------------------------|-------|---------|---------|
| Random | | | 10.83 | 4.35 | 1.3048 |
| Nullspace | | | 10.57 | 4.32 | 1.1943 |
| Nullspace | ✓ | | 11.04 | 4.77 | 1.1870 |
| Nullspace | | ✓ | 11.06 | 4.38 | 1.1870 |
| Nullspace | ✓ | ✓ | **11.57** | **4.99** | **1.1690** |

Table 4: Full ablation study on the nullspace initialization and loss terms.

$$E_{\text{prob}}(\phi) = \frac{\|\phi P_{l+1}\|_f}{\|\phi\|_f}, \qquad E_{\text{safe}}(\phi) = \frac{\|\phi N_{l+1}\|_f}{\|\phi\|_f}, \quad \forall \phi \in \{\phi_{\text{up},l}, \phi_{\text{down},l}^\top\}. \tag{13}$$

A larger Frobenius norm indicates stronger alignment of the trained adapter matrix($\phi_{\text{up},l}, \phi_{\text{down},l}$) with the corresponding subspace; in our design, the primary goal is to increase $E_{\text{safe}}$ (alignment to $N_{l+1}$).

In Table 5a, initialization markedly increases alignment to $N_{l+1}$: $E_{\text{safe}}$ reaches $0.83/0.76$ for $\phi_{\text{up},l}$ at $l=17/23$, and $0.55/0.58$ for $\phi_{\text{down},l}^\top$. Both are under $0.27$ for random initialization. This provides strong evidence that the initialization *guides optimization into the nullspace*, yielding substantially higher $E_{\text{safe}}$ across layers and for both $\phi_{\text{up},l}$ and $\phi_{\text{down},l}^\top$, which indicates successful guidance toward the nullspace directions. Meanwhile, $E_{\text{prob}}$ remains lower or comparable under SiNGER, but our objective is not to minimize $E_{\text{prob}}$ per se; rather, to ensure that the learned parameters predominantly occupy $N_{l+1}$ so as to suppress high-norm amplification while preserving useful directions. Although nullspace initialization stabilizes the refinement direction and avoids perturbations that conflict with the subsequent block, it does not lead to measurable performance improvements (Table 4).

**Loss Term.** We ablate the loss design to verify the role of information preservation. Our full objective uses both outlier suppression $\mathcal{L}_{\text{outlier}}$ and information preservation $\mathcal{L}_{\text{info}}$, whereas the ablated

| Layer | Matrix | Init | $E_{\text{prob}}\downarrow$ | $E_{\text{safe}}\uparrow$ |
|---|---|---|---|---|
| 17 | $\phi_{\text{up},l}$ | Random | 0.2565 | 0.2532 |
| 17 | $\phi_{\text{up},l}$ | SiNGER | **0.1479** | **0.8337** |
| 23 | $\phi_{\text{up},l}$ | Random | 0.2537 | 0.2541 |
| 23 | $\phi_{\text{up},l}$ | SiNGER | **0.1833** | **0.7589** |
| 17 | $\phi_{\text{down},l}^{\top}$ | Random | 0.3100 | 0.2494 |
| 17 | $\phi_{\text{down},l}^{\top}$ | SiNGER | **0.2847** | **0.5485** |
| 23 | $\phi_{\text{down},l}^{\top}$ | Random | 0.3025 | 0.2641 |
| 23 | $\phi_{\text{down},l}^{\top}$ | SiNGER | **0.2746** | **0.5774** |

(a) Initialization methods.

| Pair | $\mathcal{L}_{\text{outlier}}$ | $\mathcal{L}_{\text{info}}$ | mean $\pm$ std $\downarrow$ | median |
|---|---|---|---|---|
| $F^T \leftrightarrow \hat{F}^T$ | ✓ | | $14.22 \pm 1.45$ | 14.28 |
| $F^T \leftrightarrow \hat{F}^T$ | ✓ | ✓ | **$7.25 \pm 0.84$** | 7.19 |
| $F^T \leftrightarrow F^S$ | ✓ | | $72.36 \pm 7.61$ | 71.85 |
| $F^T \leftrightarrow F^S$ | ✓ | ✓ | **$41.71 \pm 7.01$** | 40.89 |

(b) Information preservation term.

Table 5: Ablation studies on the initialization method and the information preservation loss.

variant uses $\mathcal{L}_{\text{outlier}}$ only. (Using $\mathcal{L}_{\text{info}}$ alone admits the trivial solution $\|\Delta\|=0$ and yields no updates.)

To assess preservation of teacher information, we measure the Gram distance between $F^T$ and $\hat{F}^T$. Additionally, to evaluate the final effect on distillation, we measure how well the student features $F^S$ preserve teacher relations by comparing $F^S$ against $F^T$. Distances are computed per image and summarized over ImageNet-1K. For a feature map $F$, let $G(X) = FF^{\top}$ and define

$$D_G(F_i, F_j) = \big\| G(F_i) - G(F_j) \big\|_f.$$

In Table 5b, lower $D_G$ indicates better preservation of pairwise feature relations. Compared to $\mathcal{L}_{\text{outlier}}$ alone, adding $\mathcal{L}_{\text{info}}$ nearly halves the $D_G$ distance ($14.22 \to 7.25$) and substantially improves teacher–student alignment ($72.36 \to 41.71$). Thus, the information preservation term prevents degenerate updates and maintains the relational geometry that is crucial for effective transfer. Table 4 shows that adding $\mathcal{L}_{\text{outlier}}$ yields the largest improvement on both ImageNet-1K and ADE-20K, as it directly mitigates the dominant artifact tokens that bias distillation. $\mathcal{L}_{\text{info}}$ provides additional gains by enforcing information consistency between the teacher and student. When all components are combined, the student reaches its best performance across all tasks, demonstrating that SiNGER functions most effectively as an integrated framework rather than as a set of independent mechanisms. These results highlight the individual contribution of each component.

**Hyperparameter Sensitivity.** Two hyperparameters are required in SiNGER: $\alpha$ and $r$. $\alpha$ determines the strictness of artifact filtering by setting the percentile threshold based on the Gaussian-like distribution of the patch norms $\|F_{l,i}^{\text{T}}\|$ across $i$. $r$ controls the capacity of the perturbation $\Delta F^{\text{T}}$ applied to $F^{\text{T}}$. A larger $r$ allows the adapter to explore a wider subspace, but may distort the semantic structure of the features. Conversely, if $r$ is too small, the limited degrees of freedom restrict the adapter from effectively suppressing artifacts, while also risking the loss of informative components.

Table 6 reports the sensitivity of $\alpha$ and $r$ on NYUd-v2. We observe that performance degrades when $r$ is too small or too large, confirming the need for balanced capacity. Similarly, extreme values of $\alpha$ either under-filter artifacts or discard informative signals. The observed trends match our theoretical intuition: performance improves

| $r$ | RMSE ($\downarrow$) | $\delta_{1.25}$ ($\uparrow$) |
|---|---|---|
| 8 | 1.4545 | 33.97 |
| **16** | **1.4395** | **33.79** |
| 32 | 1.4907 | 33.07 |
| 64 | 1.6485 | 28.91 |

(a) Rank sweep with $\alpha = 0.95$.

| $\alpha$ | RMSE ($\downarrow$) | $\delta_{1.25}$ ($\uparrow$) |
|---|---|---|
| 0.90 | 1.5989 | 29.78 |
| **0.95** | **1.4395** | **33.79** |
| 0.97 | 1.4748 | 33.66 |
| 0.99 | 1.5321 | 30.80 |

(b) Quantile threshold sweep with $r = 16$.

Table 6: Rank and quantile threshold sweeps on NYUd-v2. We conduct a grid search over candidate values and select the configuration that yields the best performance.

when artifact suppression and information preservation are balanced, but deteriorates when either dominates. At the same time, the results show robustness–performance does not collapse outside the optimal point, indicating stability of the framework. Finally, the chosen hyperparameters ($r = 16$ and $\alpha = 0.95$) generalize well across other tasks and datasets, and we adopt them as the default configuration.

**Distillation Layers.** Selecting the distillation layers is critical because we aim to distill artifact-prone features. To ensure gradients traverse the entire backbone, we always distill the last layer ($l = 23$ in ViT-L). Beyond this, we select an additional intermediate layer by inspecting teacher feature trends (see Appendix H). Since our method is an artifacts-aware approach, we first pinpointed the location where artifacts occur. For ViT-L, we observed that artifacts appear after $l = 11$. We additionally

| Layers | | | NYUd-v2 |
|:---:|:---:|:---:|:---:|
| 11 | 17 | 23 | RMSE ($\downarrow$) |
| ✓ | | ✓ | 0.9554 |
| | ✓ | ✓ | **0.9406** |
| ✓ | ✓ | ✓ | 0.9624 |

Table 7: Distillation layer selection.

select the intermediate layer at $l = 17$. Across three variants, the $l = 17, 23$ configuration performs best, as shown in Table 7.

## 5 DISCUSSION

**Approximation Gap.** While we verified in Table 5a that our adapter maintains high alignment with the approximated nullspace, this internal consistency holds little value if the approximation itself fails to reflect the true non-linear nullspace. Since the ground truth nullspace is analytically intractable, we assessed the validity of our proxy by comparing our FFN-centric linearization against a full Jacobian baseline using a subset of ImageNet-1K. Specifically, we perturbed input images along the null directions computed by each method and measured the deviation between the block's original output and the output produced from these perturbed inputs.

| Metric | SiNGER | Jacobian |
|:---|:---:|:---:|
| $L_2$ ($\downarrow$) | **0.169** | 0.191 |
| Cosine sim ($\uparrow$) | **0.9787** | 0.9564 |
| CKA ($\uparrow$) | **0.9975** | 0.9947 |

Table 8: The output deviation of the non-linear block when inputs are perturbed along null directions computed by ours versus the full Jacobian.

The results in Table 8 indicate that our method induces minimal output deviations across all metrics ($L_2$ difference, Cosine similarity, and CKA), showing consistency comparable to the full Jacobian baseline. This empirically suggests that our approximation serves as a valid and robust proxy for the non-linear block.

**Complexity.** We also analyzed the computational overhead of SiNGER. The one-time SVD initialization is negligible ($< 0.2$s), and the lightweight adapters ($r = 16, d^{\mathrm{T}} = 1024$) introduce only 1.2% additional parameters (65K) to a ViT-T model. Regarding training compute, we observed an approximate 10% increase in time per epoch. It is worth noting that while the adapter operations themselves are computationally efficient, this overhead primarily stems from the extra forward pass through the teacher's subsequent block required for $\mathcal{L}_{\mathrm{info}}$.

## 6 CONCLUSION

In this work, we investigated the challenge of artifact transfer in knowledge distillation for ViTs. We showed that high-norm artifacts in teacher representations degrade interpretability and are naïvely inherited by student models, limiting the effectiveness and benefits from scaling of conventional distillation approaches. To address this issue, we proposed a distillation framework, namely SiNGER, and a nullspace-guided adapter that introduces minimal perturbations to suppress artifacts while preserving informative representations. Our framework demonstrates consistent improvements over existing methods across a diverse set of downstream tasks, yielding both higher accuracy and more interpretable features. We believe this perspective opens new directions for artifact-robust distillation and provides insights into the broader problem of transferring knowledge from over-parameterized models.

**Limitations and Future Work.** Nevertheless, our method has limitations. It suppresses artifacts rather than fully eliminating their sources. Since the goal is to retain as much teacher information as possible, the root causes of representation degradation remain. Future work will extend our approach to a wider range of foundation models and multi-modal settings, exploring whether nullspace-guided perturbations can serve as a general mechanism for reliable model compression and adaptation.

ACKNOWLEDGMENTS

This work was supported in part by the National Research Foundation of Korea (NRF) grant funded by the Korean government (MSIT) under Grant No. RS-2025-00564137, and in part by Convergence security core talent training business support program under Grant IITP-2023-RS-2023-00266615.

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

# APPENDIX

## SINGER: A CLEARER VOICE DISTILLS VISION TRANSFORMERS FURTHER

## A  TRICKS FOR CALCULATING NULLSPACE

### A.1  WEIGHTS LINEARIZATION

We generally compute a nullspace via the Singular Value Decomposition (SVD) of a linear operator $M \in \mathbb{R}^{d \times d}$, obtaining a left-nullspace $\mathcal{N} = \text{Null}(M^\top)$. This procedure presumes that the target $M$ is linear. However, a transformer block $W_l$ at layer $l$ is inherently non-linear due to attention, activation, and residual pathways, so the SVD-based nullspace of the block $W_l$ is not directly defined.

Let $x \in \mathbb{R}^{1 \times d}$ denote the row-vector feature. A standard Pre-LN transformer block at layer $l$ can be written as

$$y_l = x_l + \text{MHA}\big(\text{LN}(x_l)\big),$$
$$x_{l+1} = y_l + \text{FFN}\big(\text{LN}(y_l)\big),$$

where both $\text{MHA}(\cdot)$ and $\text{FFN}(\cdot)$ include non-linear operations (softmax attention, elementwise activations) and the residual additions further couple the sub-layers. Consequently, there is no single linear matrix $M$ that exactly represents $W_l$ for SVD, motivating a linearization that we introduce next.

To compute a nullspace for a non-linear block $W_l$, we first replace it with a linear surrogate $\tilde{W}_l$. Our key design choice is to linearize only the FFN sub-layer, motivated by an empirical study showing that the FFN induces larger relative feature changes than self-attention (SA).

We measure sub-layer-wise changes on a ViT teacher by sampling $N{=}5000$ random ImageNet training images (uniform over class folders), resizing to $224{\times}224$, normalizing, and running a forward pass to obtain per-layer tokens. For each layer $l$ and each block, we then reapply the block with instrumented intermediates:

$$x_{\text{in}} \in \mathbb{R}^{B \times (1+P) \times d},$$

$$x_{\text{SA}} = x_{\text{in}} + \text{MHA}\big(\text{LN}(x_{\text{in}})\big),$$

$$h_1 = \text{LN}(x_{\text{SA}}),$$

$$z_1 = h_1 W_1 + b_1,$$

$$a_1 = \text{GELU}(z_1),$$

$$z_2 = a_1 W_2 + b_2,$$

$$x_{\text{out}} = x_{\text{SA}} + z_2,$$

where $W_1, W_2$ are the FFN weights (expand-then-project), and we ignore the stochastic drop-path in reporting expectations. We exclude the [CLS] token and compute patch-wise $\ell_2$-norms.

For each image and layer, we aggregate over patches using the mean and record four quantities:

$$\Delta_{\text{SA}} := \text{mean}_{\text{patch}} \frac{\|x_{\text{SA}} - x_{\text{in}}\|_2}{\|x_{\text{in}}\|_2} \qquad \Delta_{\text{FFN}} := \text{mean}_{\text{patch}} \frac{\|x_{\text{out}} - x_{\text{SA}}\|_2}{\|x_{\text{SA}}\|_2}$$

$$G_{\text{FFN1}} := \text{mean}_{\text{patch}} \frac{\|a_1\|_2}{\|h_1\|_2} \qquad G_{\text{FFN2}} := \text{mean}_{\text{patch}} \frac{\|z_2\|_2}{\|a_1\|_2}$$

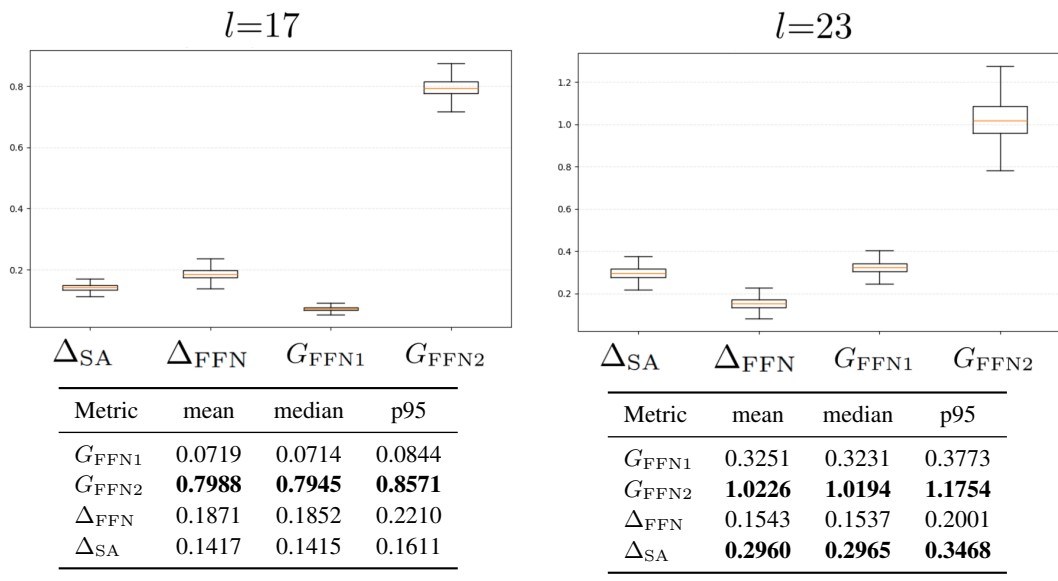

Figure 7: Sub-layer change analysis at two depths. **Top:** box-plots of relative changes/gains in each layer. **Bottom:** summary statistics in each layer.

From Figure 7, the dominant amplification occurs in the second FFN stage ($G_{\text{FFN2}}$ is largest at both depths), and the net FFN residual $\Delta_{\text{FFN}}$ is comparable to or larger than the SA residual depending on the layer. This indicates that the principal source of norm inflation lies within the FFN pathway, especially its projection stage.

Guided by this analysis, we exclude the non-linear SA pathway when constructing a linear operator for SVD and focus on the FFN inside $W_l$. Since the non-linearity enters the FFN only via the GELU between two linear maps, removing GELU (and biases) yields a linear surrogate:

$$\text{FFN}(h) \approx h\, W_{FFN1} W_{FFN2} = h\, \tilde{W} \tag{14}$$

with row-vector features and right multiplication (the column-vector convention uses $\tilde{W}^\top = W_{FFN2}^\top W_{FFN1}^\top$). We refer to $\tilde{W}_l$ as the linearized weights of block $l$.

## A.2 Nullspace of linearized weights

Now, we can compute the SVD of the linearized FFN matrix $\tilde{W}_l \in \mathbb{R}^{d \times d}$: We compute its SVD

$$\tilde{W}_l = U_l \Sigma_l V_l^\top, \qquad \Sigma_l = \text{diag}(\sigma_1 \geq \cdots \geq \sigma_d \geq 0). \tag{15}$$

A left-nullspace basis of dimension $r$ is obtained by selecting the $r$ left singular vectors associated with the $r$ smallest singular values:

$$N_l \in \mathbb{R}^{d \times r}, \qquad N_l^\top N_l = I_r, \qquad \text{cols}(N_l) = U_l^{(:,\, d-r+1:d)}. \tag{16}$$

For any row vector $v^\top \in \text{span}(N_l)$ we then have the approximate-null condition

$$v^\top \tilde{W}_l = v^\top U_l \Sigma_l V_l^\top \approx 0, \tag{17}$$

since the selected modes correspond to the smallest singular values.

A common concern is that $\tilde{W}_l$ could be numerically full rank, making the exact nullspace trivial. We therefore quantify a practical $\varepsilon$-nullspace using two diagnostics defined below, and visualize them in Figure 8.

Let the singular values of $\tilde{W}_l \in \mathbb{R}^{d \times d}$ be $\sigma_1 \geq \sigma_2 \geq \cdots \geq \sigma_d \geq 0$. Define the cumulative energy

$$E(k) := \frac{\sum_{i=1}^{k} \sigma_i^2}{\sum_{i=1}^{d} \sigma_i^2} \in [0,1],$$

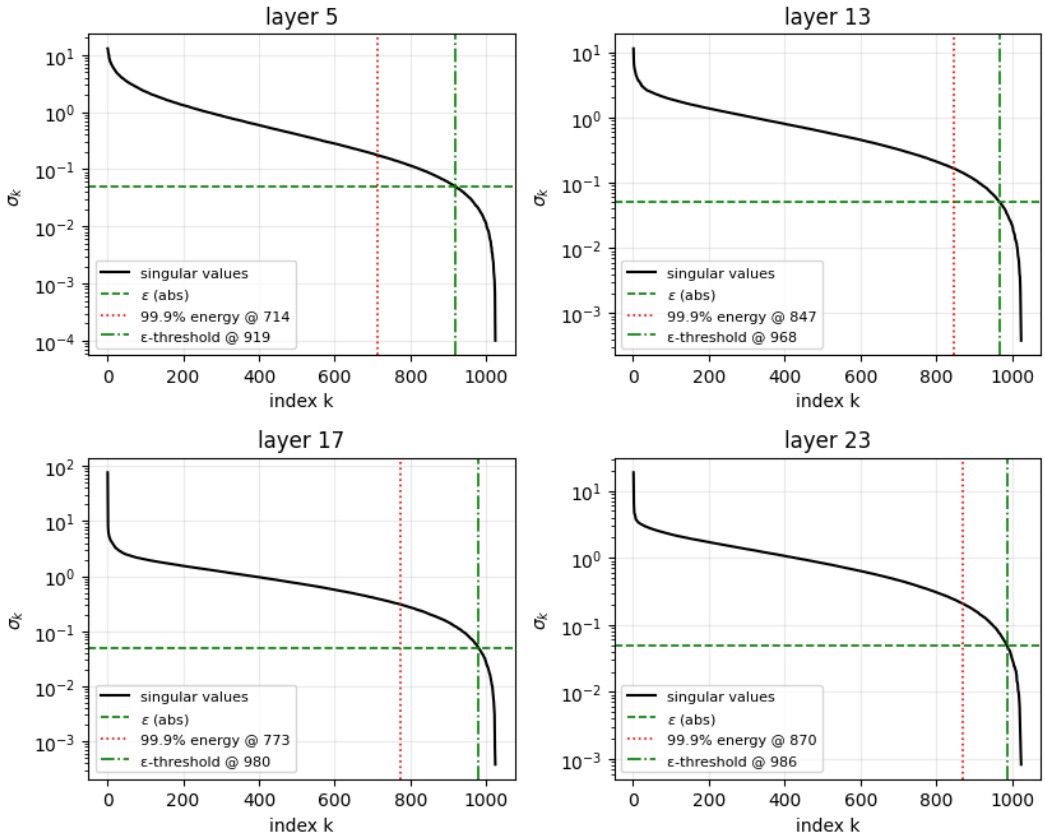

Figure 8: Singular-value spectra of $\tilde{W}_l$ for representative layers ($l = 5, 13, 17, 23$) with a green horizontal line at the absolute threshold $\varepsilon = 0.05$ and red vertical lines marking $k_{\text{energy}}$ (99.9% cumulative energy) and $k_\varepsilon$ (first index with $\sigma_k \leq \varepsilon$).

and for a target level $\rho \in (0, 1)$

$$k_{\text{energy}}(\rho) := \min\{\, k \in \{1, \ldots, d\} \, : \, E(k) \geq \rho \,\}. \tag{18}$$

For an absolute tolerance $\varepsilon > 0$, define the first crossing index

$$k_\varepsilon := \min\{\, k \in \{1, \ldots, d\} \, : \, \sigma_k \leq \varepsilon \,\}, \qquad r_\varepsilon := d - k_\varepsilon + 1, \tag{19}$$

so that the $\varepsilon$-tail has dimension $r_\varepsilon$ and is spanned by the last $r_\varepsilon$ left singular vectors.

As summarized by Figure 8, we consider a ViT-L teacher with $d{=}1024$ and focus on an intermediate block ($l{=}17$). At this layer, the cumulative-energy index is $k_{\text{energy}}(0.999){=}773$, so the low-energy tail has size $d - k_{\text{energy}}{=}251$. With an absolute tolerance $\varepsilon{=}0.05$, the first-crossing index is $k_\varepsilon{=}980$, yielding an $\varepsilon$-tail of dimension $r_\varepsilon{=}d - k_\varepsilon + 1{=}45$. Consequently, the tail span forms a high-quality *approximate nullspace*: for any $v^\top$ in this subspace,

$$\|v^\top \tilde{W}_l\|_2 \leq \varepsilon \|v\|_2,$$

which justifies nullspace-guided updates that suppress outlier energy while preserving informative structure.

## B ARCHITECTURAL SPECIFICATIONS

This section summarizes the architectural specifications of all teacher and student models used in our experiments. Table 9 lists the depth, embedding size, number of attention heads, the number of register tokens, the number of parameters, and GFLOPs for each architecture.

| Model | Depth | Embedding Size | Heads | Registers | Params (M) | GFLOPs |
|-------|-------|----------------|-------|-----------|------------|--------|
| ViT-Huge | 32 | 1280 | 16 | 0 | 630.92 | 254.63 |
| ViT-Large | 24 | 1024 | 16 | 0 | 304.33 | 123.11 |
| ViT-Base | 12 | 768 | 12 | 0 | 86.57 | 35.13 |
| ViT-Small | 12 | 384 | 6 | 0 | 22.05 | 9.20 |
| ViT-Tiny | 12 | 192 | 3 | 0 | 5.72 | 2.51 |
| DeiT-III-Large | 24 | 1024 | 16 | 0 | 304.37 | 123.11 |
| DeiT-III-Base | 12 | 768 | 12 | 0 | 86.59 | 35.13 |
| DeiT-III-Small | 12 | 576 | 6 | 0 | 22.06 | 9.20 |
| DeiT-III-Tiny | 12 | 384 | 3 | 0 | 5.72 | 2.51 |
| DINOv2-Large | 24 | 1024 | 16 | 0 | 303.35 | 123.11 |
| DINOv2-reg-Large | 24 | 1024 | 16 | 4 | 303.35 | 125.68 |
| DINOv3-Large | 24 | 1024 | 16 | 4 | 303.08 | 125.68 |

Table 9: Architectural specifications of all models used in the experiments.

All models follow their official architectures. DINOv2-reg (Darcet et al., 2024) incorporates register tokens, whereas DINOv2 (Oquab et al., 2024), DINOv2-reg, and DINOv3 (Siméoni et al., 2025) use ViT-style backbones with the same depth, hidden size, and number of heads as ViT-Large, but differ in training strategy and design details.

Across all configurations, student layers are aligned with every second teacher layer during distillation. Other architectural components (patch size, MLP ratio, and tokenization) follow the official implementations of each model family.

## C  VARIOUS TEACHER-STUDENT PAIRS

This section provides additional analysis of SiNGER under teacher configurations that differ from those used in the main paper. We evaluate three scenarios: distillation from cleaner teachers with minimal artifacts, cross-family distillation between heterogeneous architectures, and scaling the teacher from Large to Huge capacity (Table 10). All experiments were conducted on a small subset of ImageNet-1K due to computational constraints, but the training protocol was kept identical across methods for fair comparison.

| Distill | IN-1K | ADE | NYUv2 |
|---------|-------|-----|-------|
| DINOv2-reg-L → ViT-T | | | |
| FitNet | 6.97 | 3.06 | 1.3711 |
| ViTKD | 1.81 | 1.57 | 1.3499 |
| SiNGER | 6.10 | 2.78 | 1.4058 |
| DINOv3-L → ViT-T | | | |
| FitNet | 1.84 | 1.16 | 1.3868 |
| ViTKD | 0.23 | 0.42 | 1.3157 |
| SiNGER | 1.19 | 1.06 | 1.4550 |

(a) Cleaner teachers.

| Distill | IN-1K | ADE | NYUv2 |
|---------|-------|-----|-------|
| DINOv2-L → DeiT-III-T | | | |
| FitNet | 2.90 | 2.86 | 1.2328 |
| ViTKD | 2.84 | 2.54 | 1.2856 |
| SiNGER | 3.40 | 2.91 | 1.2222 |

(b) Heterogeneous families.

| Distill | IN-1K | ADE | NYUv2 |
|---------|-------|-----|-------|
| ViT-L → ViT-T | | | |
| SiNGER | 11.57 | 4.99 | 1.1690 |
| ViT-H → ViT-T | | | |
| FitNet | 9.78 | 4.75 | 1.3006 |
| SiNGER | 14.94 | 6.82 | 1.2438 |

(c) Larger teacher.

Table 10: Distillation across diverse teacher–student pairs.

**Cleaner Teachers.** To examine the behavior of SiNGER when high-norm artifacts are reduced at the teacher level, we distilled from two cleaner teacher baselines: DINOv2 with register tokens (DINOv2-reg, Darcet et al. (2024)) and DINOv3 (Siméoni et al., 2025) (Table 10a). Both exhibit substantially lower artifact magnitude in their intermediate representations.

Across these settings, SiNGER performs competitively but does not consistently surpass the strongest baseline in absolute accuracy. For DINOv2-reg → ViT-T, SiNGER improves segmentation accuracy over FitNet but falls slightly behind ViTKD on ImageNet-1K and NYUd-v2. A similar pattern is observed in DINOv3 → ViT-T, where SiNGER achieves stable but modest results while outperforming FitNet on certain tasks.

These outcomes align with the design premise of SiNGER: the method targets scenarios in which the teacher contains high-norm artifact tokens that dominate the refinement direction and bias the student. When artifacts are already minimal, the suppression loss may attenuate useful high-norm

channels, producing minor reductions in absolute performance. Nonetheless, SiNGER remains more stable than ViTKD, whose random masking strategy can behave inconsistently when artifact structure is weak or absent. This indicates that SiNGER provides a more principled refinement mechanism even in unfavorable teacher conditions.

**Heterogeneous Families.** To assess generality beyond same-family ViTs, we conducted cross-family distillation from DINOv2-L to DeiT-III-T (Table 10b). Owing to the reduced dataset size, absolute performance is lower than full-scale training; however, the relative gains are significant. SiNGER improves over FitNet by 17.2% on ImageNet-1K validation, 1.75% mIoU on ADE-20K, and achieves a 0.86% reduction in RMSE on NYUd-v2. These results demonstrate that SiNGER effectively transfers information even when teacher and student architectures differ substantially, suggesting broad applicability of the refinement strategy.

**Scaling the Teacher.** We further evaluated whether SiNGER maintains its benefit when increasing teacher capacity. Using ViT-H (huge) as the teacher and ViT-T as the student, SiNGER improves top-1 accuracy by 5.16%p, increases mIoU by 2.07%p, and reduces RMSE compared to FitNet. Similar gains appear when distilling from ViT-L. These results confirm that the nullspace-guided refinement process remains effective for high-capacity teachers, even under constrained training budgets.

## D  FAILURE OF ViTKD

In this section, we discuss the failure of ViTKD (Yang et al., 2024) in learning the teacher representation. The core strategy behind ViTKD is masking and generation. Different from SiNGER, ViTKD does not adaptively detect artifacts and randomly discards patches regardless of their semantic validity, and generates through convolution, utilizing learnable generative tokens. This ensures students learn artifact-free representation. However, it also results in a blurred representation, which is one of the expected trivial solutions to minimize the mean squared error of the generated features, resulting in a significantly degraded representation.

As depicted in Figure 2, ViTKD successfully mimics the teacher's representation in terms of cosine similarity, but fails in building the informatively structured feature map. This structural degradation makes the representation blurry, resulting in poor downstream task adaptation.

## E  VISUALIZING DISTILLATION OUTPUTS ON A SINGLE IMAGE

We visualize a single sample at token resolution $14 \times 14$ (patches only; [CLS] excluded). For the teacher(ViT-large), we show odd-numbered blocks $l \in \{1, 3, \ldots, 23\}$. For the student(ViT-Tiny), we show layers $i \in \{0, \ldots, 11\}$ aligned with the teacher columns. In this sample, the SiNGER adapter is applied only at $l \in \{17, 23\}$.

As shown in Figure 9, the first row $F_l^{\mathrm{T}}$ exhibits artifacts: high-norm becomes more pronounced at deeper blocks. After applying SiNGER adapter, the second row $\hat{F}_l^{\mathrm{T}}$ attenuates these artifacts while preserving the informational structure, producing a more transfer-friendly teacher target. The third row visualizes the residual $\Delta F_l^{\mathrm{T}} = \hat{F}_l^{\mathrm{T}} - F_l^{\mathrm{T}}$, confirming that SiNGER removes a small set of outlier's magnitudes. Finally, the fourth row $F_l^{\mathrm{S}}$ aligns more closely with $\hat{F}_l^{\mathrm{T}}$ than with $F_l^{\mathrm{T}}$, indicating that the student learns the SiNGER-refined, structure-preserving representation rather than the original outlier-dominated one.

## F  VISUALIZATION OF OUTLIER SUPPRESSION

This appendix illustrates, on a single sample, how outlier suppression operates numerically. As shown in Figure 10, in the last layer, the patchwise norm map $\|F_l^{\mathrm{T}}\|$ contains an outlier patch with a maximum norm of $638$. At the same spatial location, the suppressed map $\|\hat{F}_l^{\mathrm{T}}\|$ drops to $54.1$. Finally, the distribution of $\|\hat{F}_l^{\mathrm{T}}\|$ appears much more uniform across patches, indicating that extremely high-norm outliers have been attenuated while the overall scale has been regularized.

## G   VISUALIZATION OF INFORMATION PRESERVATION

We confirm whether the information is actually preserved right after the $l + 1$-th layer. The cross-similarity map between $F_{l+1}^{\mathrm{T}}$ and $\hat{F}_{l+1}^{\mathrm{T}}$ are visualized in Figure 11. For each $1 \times 2$ cells, the left one shows $F_{l+1}^{\mathrm{T}} \rightarrow \hat{F}_{l+1}^{\mathrm{T}}$ similarity map, and the other one shows the contrary. As shown, in both directions, the similarity maps are almost identical. This implies the information is actually preserved even after passing the next layer, which is the source of nullspace used for initializing the proposed adapter.

## H   STUDENT VISUALIZATION

We evaluate the quality of our SiNGER-distilled feature by visualizing the similarity map (Figure 12). For each $2 \times 2$ cell, the top row is the student, the bottom row is the teacher. The left column is the norm map of the feature map, and the right column is the similarity map to the '$\times$' marked patch. The norms of the teacher's feature maps are artifact-prone, but still produce the similarity map, which semantically makes sense. The student produces artifact-suppressed feature maps while maintaining the semantic relation among the patches. This emphasizes that SiNGER effectively optimizes two objective functions, distills high-quality feature representation.

## I   CORRELATION OF OUTLIER WITH PRINCIPAL AND NULL BASES

We conduct this experiment to validate the core design choice behind our method: we perturb features along a nullspace to preserve information, but such a perturbation is only meaningful if outliers do not primarily reside in the nullspace. Otherwise, nullspace-directed updates would fail to suppress outliers. To test this, we build on Appendix A.1 and Appendix A.2: from the linearized FFN matrix $\tilde{W}_l$, we take the $r$ left singular vectors with the largest singular values as the **principal basis**, and the $r$ with the smallest singular values as the **null basis**.

Fix a layer $l$ and an image, and let $X \in \mathbb{R}^{(1+P) \times d}$ be the teacher tokens (CLS excluded below). Compute

$$\tilde{W}_l = U_l \, \Sigma_l \, V_l^{\top}, \quad U_l = [u_1, \ldots, u_d], \, \Sigma_l = \mathrm{diag}(\sigma_1 \geq \cdots \geq \sigma_d \geq 0).$$

Define the two $r$-dimensional bases

$$U_{\mathrm{prin}} = [u_1, \ldots, u_r], \qquad U_{\mathrm{null}} = [u_{d-r+1}, \ldots, u_d].$$

Let $x_p \in \mathbb{R}^{1 \times d}$ be the $p$-th patch feature with norm $n_p = \|x_p\|_2$. For a chosen $r$-dimensional orthonormal basis $U = \begin{bmatrix} u_{i_1}, \ldots, u_{i_r} \end{bmatrix} \in \mathbb{R}^{d \times r}$ (e.g., columns selected from the left singular vectors of $\tilde{W}_l$), define the normalized subspace energy.

$$E_U(p) \;=\; \frac{\displaystyle\sum_{k=1}^{r} \langle x_p, u_{i_k} \rangle^2}{\|x_p\|_2^2 + \varepsilon}$$

In words, $E_U(p)$ is the fraction of the patch's total energy captured by the subspace $U$–i.e., a norm-invariant measure of how strongly $x_p$ aligns with $U$. In our analysis, we instantiate $U$ by either of the above bases, i.e.,

$$U \in \big\{ \, U_{\mathrm{prin}}, \; U_{\mathrm{null}} \, \big\}.$$

In Figure 13, across layers we observe distinct behaviors as the rank $r$ increases. At the intermediate layer ($l$=17), outlier patches (high $\|F_l^{\mathrm{T}}\|_2$) exhibit growing values in $E_{U_{\mathrm{null}}}$ as $r$ increases, while $E_{U_{\mathrm{prin}}}$ over those same patches does not grow accordingly. Conversely, at the last layer ($l$=23), outlier patches show increasing $E_{U_{\mathrm{prin}}}$ with $r$, whereas $E_{U_{\mathrm{null}}}$ over outliers does not increase in the same manner. We quantify these patterns in Figure 13 (m),(n). For $l$=17, the patch norm correlates positively with the null subspace energy ($\mathrm{corr}(\|F_l^{\mathrm{T}}\|_2, E_{U_{\mathrm{null}}}) > 0$) and negatively with

the principal subspace energy $(\mathrm{corr}(\|F_l^{\mathrm{T}}\|_2, E_{U_{\mathrm{prin}}}) < 0)$. In contrast, for $l{=}23$ the signs flip: $\mathrm{corr}(\|F_l^{\mathrm{T}}\|_2, E_{U_{\mathrm{null}}}) < 0$ and $\mathrm{corr}(\|F_l^{\mathrm{T}}\|_2, E_{U_{\mathrm{prin}}}) > 0$.

These results indicate that at intermediate depth (e.g., $l{=}17$) the high-norm (outlier) content lies relatively closer to the null subspace, whereas at the final depth (e.g., $l{=}23$) it aligns more with the principal subspace. Accordingly, initializing updates along the nullspace at intermediate layers achieves information preservation (by construction) while still enabling outlier suppression, consistent with our design objective.

## J  DISTILLED MODELS IN LONG-TAIL LEARNING

To diagnose the source of performance degradation on iNaturalist2019, we examined the entropy of the teacher's logits across the full set, the majority classes, and the long-tail classes. The results are summarized in Table 11.

| Model | Full (53,649) | | Major (1,873) | | Long-tail (1,819) | |
|---|---|---|---|---|---|---|
| | Ent. | Acc. | Ent. | Acc. | Ent. | Acc. |
| ViT-L | 6.6336 | 71.41 | 6.6290 | 74.21 | 6.6526 | 61.24 |
| ViT-T | 6.6435 | 44.71 | 6.6421 | 48.10 | 6.6548 | 32.22 |
| SiNGER | 6.6940 | 41.08 | 6.6933 | 48.37 | 6.6989 | 24.96 |
| FitNet | 6.7097 | 39.98 | 6.7078 | 44.69 | 6.7135 | 26.50 |

Table 11: Entropy analysis in long-tail learning. The numbers in the parenthesis refer to the number of samples. Ent. and Acc. means entropy and accuracy, respectively.

A consistent pattern emerges: the teacher (ViT-L) exhibits the largest entropy increase specifically in long-tail classes, indicating substantially lower confidence and greater prediction ambiguity for minority categories. Because knowledge distillation transfers the teacher's class-level certainty, this elevated uncertainty is directly inherited by the student.

Crucially, SiNGER focuses on suppressing spatial artifact tokens but does not alter the teacher's class logits. Therefore, when the teacher already provides ambiguous supervision for rare classes, the long-tail accuracy becomes limited not by artifact noise but by the teacher's intrinsic uncertainty. This explains why the SiNGER-trained student shows smaller gains–and occasionally lower accuracy–than a ViT-T trained without distillation, despite improving performance in other scenarios.

Still, SiNGER demonstrates lower long-tail entropy than FitNet (6.6989 vs. 6.7135), suggesting that stabilizing the refinement direction yields more reliable supervision even when the teacher's uncertainty dominates.

## K  IMPLEMENTATION DETAILS

In this section, we report our implementation details for reproducibility. We trained our model on Ubuntu 22.04.5 LTS with CUDA v12.6.85 using eight NVIDIA GeForce RTX 3090 GPUs. Mixed precision training (FP16) was enabled to reduce memory consumption and accelerate computation. All experiments were conducted with a global batch size of 512, distributed evenly across GPUs using PyTorch's DistributedDataParallel (DDP).

For distillation, the 8th layer and 17th layer were selected as the distillation layers for Tiny ViT and Large ViT, respectively. We tuned the weights for the losses equally to 1.0. The model and adapters were optimized using the AdamW optimizer with a cosine annealing learning rate scheduler of a single cycle. The learning rate was initialized at $10^{-4}$ and decayed to $10^{-8}$ over 100 epochs. Weight decay was set to 0.05, and gradient clipping with a max norm of 1.0 was applied to stabilize training. Data augmentation followed common practice for ImageNet training, including random resized cropping and horizontal flipping. Input images were resized to $224 \times 224$ unless otherwise specified. During evaluation, a center crop was used. All hyperparameters and code will be released upon publication.

## L    THE USE OF LARGE LANGUAGE MODELS

As per the ICLR 2026 guidelines on the use of Large Language Models (LLMs), we disclose that an LLM was used for minor grammar corrections and polishing of the text to enhance readability, as well as for searching related research to broaden the scope of the literature review. The LLM did not contribute to the research ideation, methodology, or core findings of the paper.

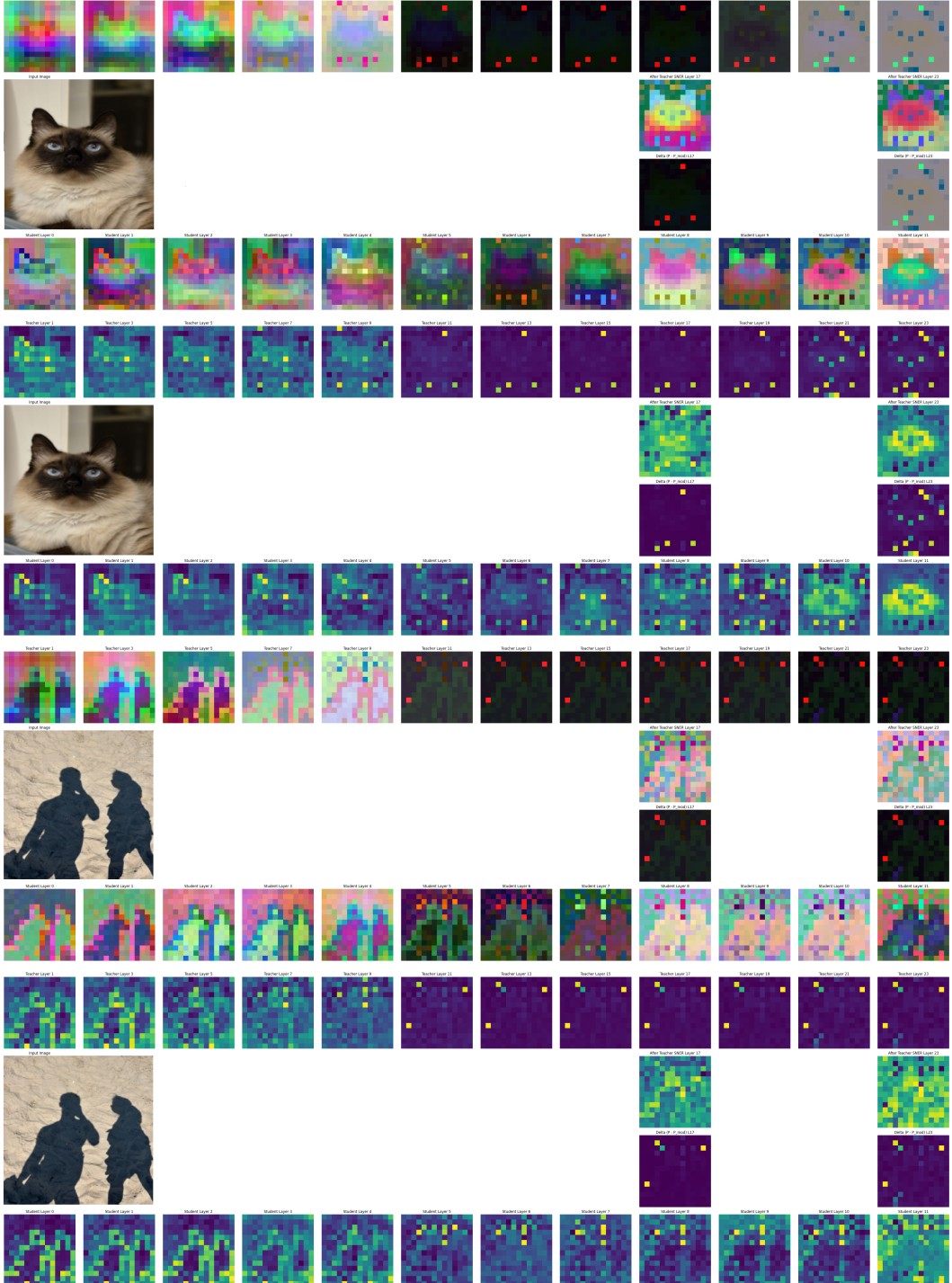

Figure 9: Distillation visualization across stages using complementary views. Each panel renders patch features either as a directional view (PCA with 3 components) or as a magnitude view (patch-wise $\ell_2$-norm). Within each panel, rows depict (top to bottom) $F_l^{\mathrm{T}}, \hat{F}_l^{\mathrm{T}}, \Delta F_l^{\mathrm{T}}, F_l^{\mathrm{S}}$.

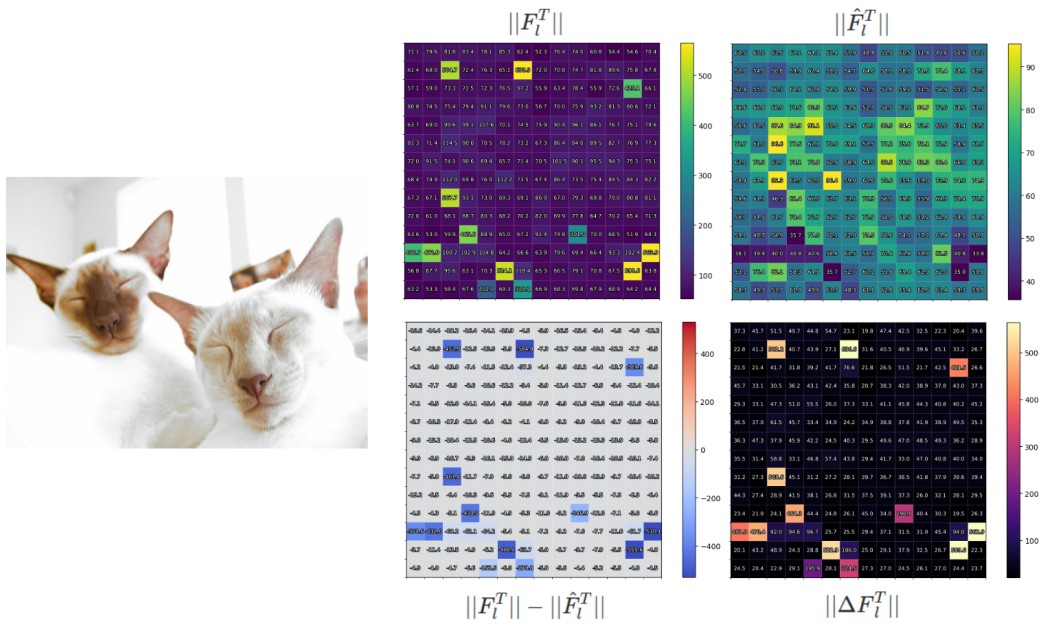

Figure 10: Left: input image. Right: patchwise visualizations. **Row 1:** $\|F_l^{\mathrm{T}}\|$ and $\|\hat{F}_l^{\mathrm{T}}\|$. **Row 2:** $\|F_l^{\mathrm{T}}\|$, $\hat{F}_l^{\mathrm{T}}$ (signed map), and $\|\Delta F_l^{\mathrm{T}}\|$ with $\Delta F_l^{\mathrm{T}} = F_l^{\mathrm{T}} - \hat{F}_l^{\mathrm{T}}$.

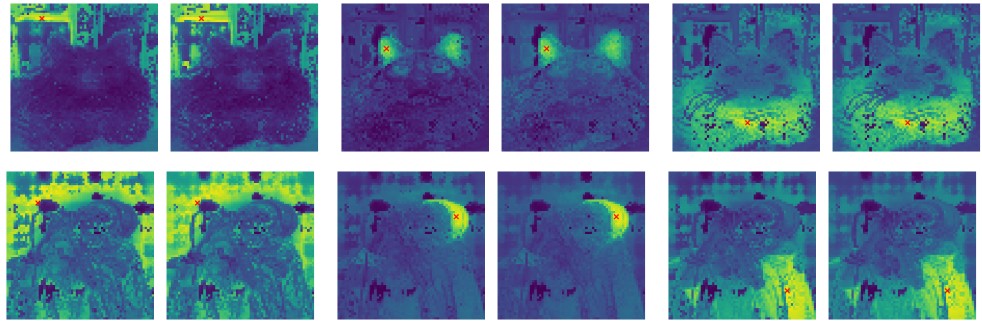

Figure 11: The cross-similarity map between $F_{l+1}^{\mathrm{T}}$ and $\hat{F}_{l+1}^{\mathrm{T}}$ to '×' mark is visualized.

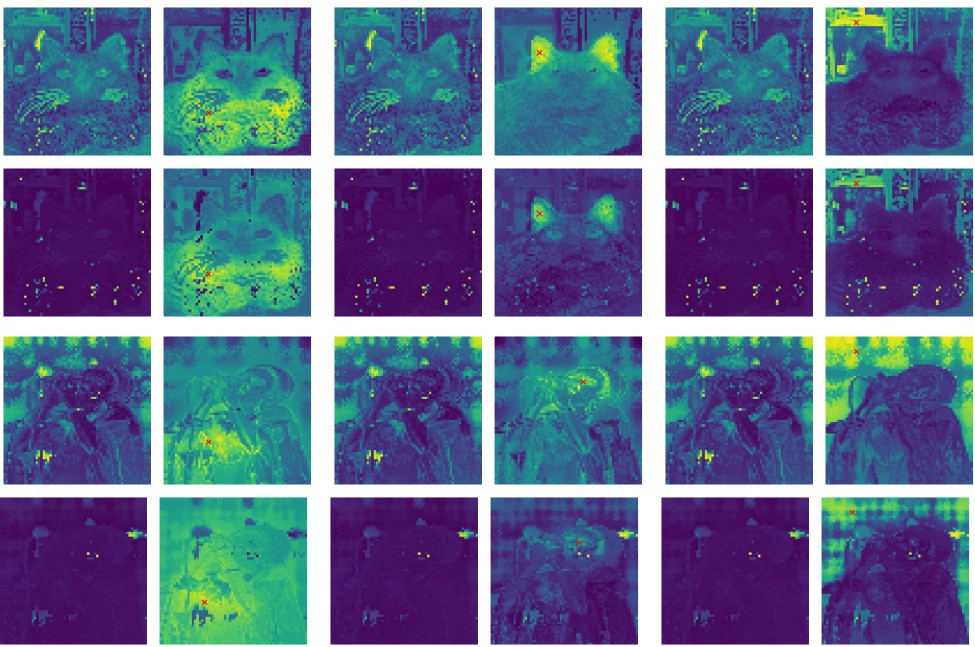

Figure 12: The similarity map to '×' mark is visualized.

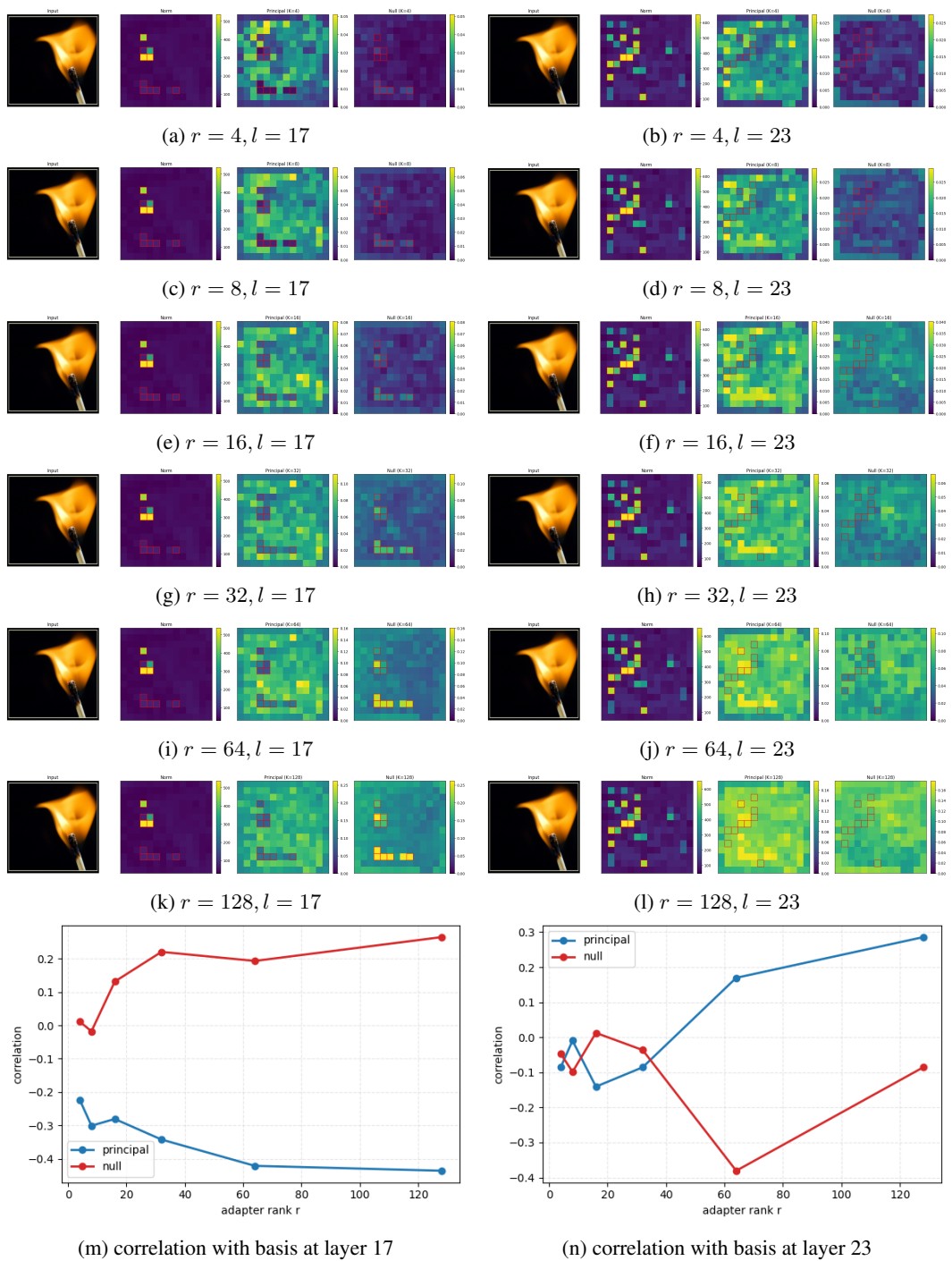

Figure 13: panels (a)–(l) show, for each setting, four views: the input image, the patchwise norm $\|F_l^{\mathrm{T}}\|_2$, and the subspace energies $E_{U_{\mathrm{prin}}}$ and $E_{U_{\mathrm{null}}}$. Panels (m)–(n) show correlation with each basis.

