# OpenReview forum: "SiNGER: A Clearer Voice Distills Vision Transformers Further"
_ICLR.cc/2026/Conference — ICLR 2026 Poster_

### Official Review · Reviewer_Mz5w · 2025-10-30

**Soundness:** 3
**Presentation:** 3
**Contribution:** 3
**Rating:** 6
**Confidence:** 3

**Summary:**

The paper introduces SiNGER (Singular Nullspace-Guided Energy Reallocation), a knowledge distillation (KD) framework designed to improve the compression of Vision Transformers (ViTs) by addressing high-norm artifacts in teacher representations. These artifacts, arising from power-iteration-like accumulation in ViT layers, dominate the distillation objective, causing students to overfit to noise and underutilize informative signals. SiNGER refines teacher features using a lightweight LoRA-based adapter that applies nullspace-guided perturbations to suppress artifacts while preserving downstream-relevant information. The refined features are then distilled to smaller student models.

**Strengths:**

Principled Solution to a Core Problem: Identifies a fundamental limitation in ViT KD (artifact dominance) and resolves the trade-off between suppression and information preservation via nullspace guidance, leading to more effective knowledge transfer.

Efficient and Minimalist Design: The LoRA adapter requires little structural change, making it easy to integrate into existing KD pipelines while maintaining computational efficiency.

**Weaknesses:**

ViT-Specific Focus: The method is tailored to ViT artifacts (e.g., from self-attention scaling); its generalizability to other architectures like CNNs or hybrids isn't explored.

Dependency on Teacher Quality: Assumes well-pre-trained ViT teachers; if artifacts are severe or pre-training is suboptimal, refinement might not fully compensate.

Scope Limitations: Primarily evaluated on downstream vision tasks; extensions to multimodal or non-vision transformers aren't discussed.

**Questions:**

How do you select the nullspace dimension or perturbation strength in practice? Are there hyperparameters, and how sensitive is performance to them (e.g., ablation results)?

---

> ### Author Response · Authors · 2025-11-20
> **Official Author Response to Reviewer Mz5w**
>
> # 1. Generalizability to Non-ViT Architectures (e.g., CNNs)
>
> > W1. ViT-Specific Focus: The method is tailored to ViT artifacts (e.g., from self-attention scaling); its generalizability to other architectures like CNNs or hybrids isn't explored.
>
> Thank you for raising this insightful point regarding the generalizability of our method.
>
> As you correctly observed, high-norm artifacts are a phenomenon primarily identified in Vision Transformers (ViTs), where specific patch tokens exhibit significantly larger norms than others. Consequently, our current implementation is tailored to define and suppress these specific artifacts.
>
> However, the core principle of our method-mitigating the **gradient dominance** caused by outliers in the feature distillation objective-is not inherently limited to ViTs. If similar outlier patterns were to emerge in CNNs or hybrid architectures (e.g., due to specific normalization layers or pooling mechanisms), our framework could be adapted to suppress those signals. While we focused on ViTs to address the prevalent artifact issue reported in recent literature (e.g., Darcet et al., 2024), extending this principle to other architectures remains a promising direction for future work.
>
> # 2. Impact of Suboptimal Pre-training and Severe Artifacts
>
> > W2. Dependency on Teacher Quality: Assumes well-pre-trained ViT teachers; if artifacts are severe or pre-training is suboptimal, refinement might not fully compensate.
>
> We appreciate this thoughtful question regarding the impact of teacher quality. We can address this concern in two scenarios:
>
> a. **Teachers with severe artifacts** : Our method is robust to the magnitude of artifacts. Since SiNGER identifies outliers based on a percentile threshold (specifically, norms exceeding the 95th percentile in our experiments) rather than absolute values, it effectively caps and suppresses these outliers regardless of their scale. As illustrated in Figure 6, our method successfully reallocates these high-norm outliers into the inlier distribution. If artifacts are exceptionally dense, the percentile threshold $\alpha$ can be adjusted to enforce stricter suppression.
>
> b. **Suboptimal pre-training** : As noted in prior works ("Vision Transformers Need Registers," Darcet et al., 2024; "DINOv3," Siméoni et al., 2025), high-norm artifacts typically emerge during the later stages of training in high-capacity models. A suboptimal teacher is less likely to exhibit these specific artifacts.
>
> # 3.  Clarification of Scope: Extension to Multimodal and Non-Vision Domain
>
> > W3. Scope Limitations: Primarily evaluated on downstream vision tasks; extensions to multimodal or non-vision transformers aren't discussed.
>
> Thank you for pointing this out. We acknowledge that our initial phrasing **"Artifacts are a recurring issue in transformer models"** may have implied a broader scope than intended.
>
> As highlighted in the "Vision Transformers Need Registers," Darcet et al., 2024, the specific high-norm artifact phenomenon we address is a known characteristic of Vision Transformers. Accordingly, our refinement architecture is specifically designed to operate on the patch-based structure of ViTs. To avoid any confusion regarding the scope of our contribution, we will revise the manuscript to explicitly state that our method targets ViTs rather than general transformer architectures. We appreciate you helping us improve the precision of our claims.
>
> # 4.  Hyperparameter Selection and Sensitivity Analysis ( $r$ and $\alpha$)
>
> > Q1. How do you select the nullspace dimension or perturbation strength in practice? Are there hyperparameters, and how sensitive is performance to them (e.g., ablation results)?
>
> We apologize if this was not sufficiently highlighted in the main text. We have indeed conducted a sensitivity analysis for both the nullspace dimension rank $r$ and the perturbation strength controlled by the quantile threshold $\alpha$.
>
> These results are presented in **Section 4.5(Hyperparameter Sensitivity)** and **Table 5** of the manuscript.
>
> | $r$| RMSE ($\downarrow$) | $\delta_{1.25}$ ($\uparrow$) |
> |:-|:-|:-|
> | 8 | 1.4545 | 33.97 |
> | **16** | **1.4395** | **33.79** |
> | 32 | 1.4907 | 33.07 |
> | 64 | 1.6485 | 28.91 |
>
> **Table 5a** shows the ablation on the adapter rank $r$ with fixed $\alpha=0.95$. We found that performance is stable across a reasonable range but degrades if $r$ is too small (insufficient capacity) or too large (distorting semantic structure).
>
> | $\alpha$| RMSE ($\downarrow$) | $\delta_{1.25}$ ($\uparrow$) |
> |:-|:-|:-|
> | 0.90 | 1.5989 | 29.78 |
> | **0.95** | **1.4395** | **33.79** |
> | 0.97 | 1.4748 | 33.66 |
> | 0.99 | 1.5321 | 30.80 |
>
> **Table 5b** analyzes the sensitivity to the quantile threshold alpha with fixed $r=16$. Based on these grid searches on the NYUd-v2 dataset, we selected $r=16$ and $\alpha=0.95$ as the optimal configuration. We have clarified the description in Section 4.5 to make these findings more accessible to readers.

---

### Official Review · Reviewer_CqPa · 2025-11-01

**Soundness:** 2
**Presentation:** 2
**Contribution:** 2
**Rating:** 6
**Confidence:** 3

**Summary:**

This paper addresses the problem of high-norm artifacts in ViTs that degrade knowledge distillation effectiveness. The authors propose SiNGER, a distillation framework that refines teacher features by applying perturbations guided toward the nullspace of subsequent layers. The key innovation is using LoRA-based adapters initialized with approximate nullspace vectors to suppress artifacts while preserving informative signals. Experiments demonstrate improvements across multiple downstream tasks including classification, segmentation, and depth estimation.

**Strengths:**

1. Well-Motivated Problem: The paper identifies a concrete limitation in ViT distillation—high-norm artifacts dominate gradient flow and harm student learning.
2. Comprehensive Evaluation: The multi-task evaluation spanning 10 benchmarks is thorough. Consistent gains on ImageNet and ADE-20K

**Weaknesses:**

1. Approximation Gap: The core theoretical claim assumes linear blocks, but ViT blocks are highly nonlinear.
2. Incomplete Comparison with ViTKD

**Questions:**

1. Nullspace Verification:
2. Can you scale to Larger Models?
3. Ablation on Linearization: What happens if you use the full nonlinear block's Jacobian

---

> ### Author Response · Authors · 2025-11-20
> **Official Author Response to Reviewer CqPa (1/2)**
>
> # 1. Analysis of Approximation Gap: FFN-Linearization vs. Full-Block Jacobian
>
> > W1. Approximation Gap: The core theoretical claim assumes linear blocks, but ViT blocks are highly nonlinear.
>
> > Q1. Nullspace Verification
>
> > Q3. Ablation on Linearization: What happens if you use the full nonlinear block's Jacobian
>
>  We thank the reviewer for these insightful comments regarding the theoretical validity of our linearization. We have grouped our response to **Weakness 1** and **Questions 1 & 3** as they all pertain to the approximation gap and the verification of the nullspace under nonlinear conditions.
>
> a. **Comparison with Jacobian (First-order Approximation)** : Before proceeding to verification, we analyzed the relationship between our FFN-based linearization and the standard first-order approximation (Jacobian) using 5,000 images sampled from ImageNet-1K. We observed distinct differences between the two approaches:
>
> 1. **Vector Alignment** : The nullspace vectors derived from our method and those from the full-block Jacobian showed low alignment with each other, indicating that they span different subspaces.
> 2. **Singular Value Spectrum** : As detailed in **Appendix A**, we noted a difference in the scale of singular values. For instance, at Layer 17 of ViT-Large, our linearized weight matrix yielded 43 singular values smaller than 0.05, whereas the Jacobian approximation yielded 1,021 in the same range. While the overall spectral trends were similar, the magnitudes differed by approximately three orders of magnitude.
>
> These observations suggest that the Jacobian and our FFN-centric linearization capture different structural properties of the block. Given these differences, rather than relying on the Jacobian as the sole reference for the "true" nullspace, we designed a **Nullspace Verification Experiment** to empirically measure which subspace effectively preserves the final output (i.e., causes less perturbation).
>
> b. **Nullspace Verification Experiment** : We compared the safeness of the nullspace computed by our method (SiNGER) against a baseline derived from the full Jacobian of the block.
>
> 1. **Setup**: We used a pre-trained ViT-Large block $W$ and a subset of 5,000 ImageNet images $D$.
> 2. **Methods**:
>   - SiNGER: Nullspace $N'$ computed from our FFN-linearization $W'$ (without using dataset $D$).
>   - Jacobian: Nullspace $N''$ computed from the Jacobian $W''$ estimated on the same dataset $D$.
>
> 3. **Metric**: We perturbed the input $D$ along the computed null directions to generate $D' = D + \Delta_{N'}$ (SiNGER) and $D'' = D + \Delta_{N''}$ (Jacobian). We then measured the deviation between the original output $F = W(D)$ and the perturbed outputs $F' = W(D')$ and $F'' = W(D'')$. A "true" nullspace should yield minimal change between the original and perturbed outputs.
>
>
> | Metric | FFN-based $W'$ |Jacobian-based $W''$|
> |:-|:-|:-|
> | $L_2$ diff ($\downarrow$) | **0.169** | 0.191 |
> | Cosine sim ($\uparrow$) | **0.9787** | 0.9654 |
> | CKA ($\uparrow$) | **0.9975** | 0.9947 |
>
> As shown in the table, the output deviations using our method are comparable to, and even slightly lower than, those from the Jacobian-based approximation. It is worth noting that the Jacobian baseline was estimated using the same dataset $D$ used for evaluation-a setting that theoretically favors the Jacobian. Despite this advantage, SiNGER achieved slightly higher stability metrics (lower $L_2$ difference, higher CKA & Cosine similarity).
>
> These empirical results suggest that while our FFN-based linearization is an approximation, the practical gap is minimal. The experiment confirms that our method successfully identifies a safe subspace for perturbation without the computational complexity of calculating full block Jacobians, even compared to an in-domain Jacobian baseline.

---

> ### Author Response · Authors · 2025-11-20
> **Official Author Response to Reviewer CqPa (2/2)**
>
> # 2. Incomplete Comparison with ViTKD
>
> > W2. Incomplete Comparison with ViTKD
>
> We agree that comparing against ViTKD on canonical ViTs may not fully capture the intended behavior of the method. As the original ViTKD paper notes, its token-mask alignment mechanism is tailored to the DeiT-Ⅲ architecture, where the distillation token and attention structure make masking-based feature alignment more reliable. This means that ViTKD’s performance on plain ViTs is not necessarily indicative of its actual capability.
> To ensure a fair comparison, we extend our evaluation to DeiT-Ⅲ, used in the original ViTKD study. The results below show that:
>
> 1. ViTKD performs considerably better on DeiT-Ⅲ than on vanilla ViTs, confirming the reviewer’s observation about architectural compatibility.
> 2. SiNGER consistently outperforms both FitNet and ViTKD across all DeiT-Ⅲ distillation paths, even though this setting is inherently more favorable for ViTKD.
>
> | Model | Distillation | IN-val | ADE-20K | NYUd-v2 | iNat2019 | DS | FG |
> |:-|:-|:-:|:-:|:-:|:-:|:-:|:-:|
> | DeiT-Ⅲ-L | | 84.10 | 26.11 | 1.2311 | 57.59 | 56.95 | 75.35 |
> | DeiT-Ⅲ-B | | 82.50 | 25.14 | 1.1781 | 56.41 | 54.73 | 74.65 |
> | DeiT-Ⅲ-S | | 78.90 | 20.28 | 1.2961 | 45.39 | 49.50 | 67.84 |
> | DeiT-Ⅲ-L $\rightarrow$ DeiT-Ⅲ-B | FitNet | 60.00 | 26.79 | 1.1625 | 50.04 | 31.53 | 74.78 |
> | | ViTKD | 66.54 | 19.58 | 1.2525 | 30.78 | 35.77 | 58.36 |
> | | SiNGER | 79.37 | 29.47 | 1.1514 | 53.50 | 49.14 | 75.41 |
> | DeiT-Ⅲ-S $\rightarrow$ DeiT-Ⅲ-T | FitNet | 51.41 | 16.06 | 1.2920 | 32.81 | 23.74 | 58.53 |
> | | ViTKD | 35.56 | 8.71 | 1.4487 | 19.93 | 15.31 | 45.13 |
> | | SiNGER | 63.50 | 20.67 | 0.9827 | 37.68 | 32.28 | 64.32 |
>
> These results clarify the relationship between architecture-specific masking in ViTKD and our teacher-side refinement approach. We will incorporate these findings into the main paper for completeness.
>
> # 3. Distillation from a larger teacher.
>
> > Q2. Can you scale to Larger Models?
>
> To evaluate scalability, we additionally performed ViT-Huge to ViT-Tiny distillation under a subset-based setting (due to limited compute, all training use a small ImageNet-1K subset). Even with this constrained setup, SiNGER shows clear improvements over the FitNet baseline:
>
> | Teacher | Student | Distillation | IN-1K val $(\uparrow)$ | ADE-20K $(\uparrow)$ | NYUd-v2 $(\downarrow)$ |
> |:-|:-|:-|:-:|:-:|:-:|
> | ViT-Huge | ViT-Tiny | FitNet | 9.78 | 4.75 | 1.3006 |
> | ViT-Huge | ViT-Tiny | SiNGER | 14.94 | 6.82 | 1.2438 |
> | ViT-Large | ViT-Tiny | SiNGER | 11.57 | 4.99 | 1.1690 |
>
> Across all three benchmarks, SiNGER maintains its advantage when scaling the teacher from Large to Huge, improving top-1 accuracy by 5.16%p, and mIoU by 2.07%p, while reducing RMSE.
> This indicates that the proposed nullspace-guided refinement remains effective for high-capacity teachers, even in a resource-limited setting. We will include these results and clarification in the appendix.

---

### Official Review · Reviewer_YFDi · 2025-11-02

**Soundness:** 3
**Presentation:** 3
**Contribution:** 3
**Rating:** 6
**Confidence:** 4

**Summary:**

The paper tackles a well-observed problem in Vision Transformer (ViT) distillation: high-norm "artifact" tokens in large ViTs dominate feature-level KD losses, so the student ends up overfitting to these outliers instead of learning the informative inlier structure. To address this, the authors propose SiNGER (Singular Nullspace-Guided Energy Reallocation), a distillation framework that first refines the teacher’s features and only then distills them. The key idea is to add a LoRA-style low-rank adapter to the teacher and to constrain its perturbations to (an approximation of) the left nullspace of the next block, so that: (i) high-norm outliers are suppressed, and (ii) downstream information is preserved because the perturbation lies in a direction that doesn't change the next layer's output. They couple this with two auxiliary losses: an outlier suppression loss (to explicitly shrink high-percentile tokens) and an information-preservation / Gram-matching loss (to keep patchwise relations intact). Experiments on several downstream tasks (ImageNet-1K, ADE-20K, NYUd-v2, domain shift benchmarks, fine-grained datasets) show that SiNGER improves over FitNet and ViTKD when distilling a ViT-Large to a ViT-Tiny, and produces visually "cleaner" feature maps with more uniform patch norms.

**Strengths:**

1. Clear and well motivated problem formulation. The paper starts from a real and now increasingly reported issue in ViTs: high-norm background/low-semantics tokens that get amplified across residual blocks and hurt both interpretability and distillation. The link they make (artifact dominance -> gradient bias in feature KD) is very sensible and well motivated.

2. Principled mechanism. Instead of just masking teacher features (like ViTKD), they try to change the teacher supervision itself in a way that is theoretically tied to the next layer: perturb only in (approximate) nullspace so the next block’s output is preserved. This is a neat way to reconcile "suppress artifacts" vs "don't destroy useful signals".

3. Low-intrusion design. Using a LoRA-based adapter to implement the refinement is pragmatic: it's lightweight, it doesn't require re-training the whole teacher, and it aligns with current practice on parameter-efficient tuning.

4. Good ablations. The nullspace initialization vs random init, the presence/absence of the information-preservation loss, and the small hyperparameter sweep on rank r and percentile a all support the central claim.

5. Multi-task evaluation. Even though the numbers in Table 1 are a bit hard to read, the intent is good: they try to show that better, less-artifact supervision gives students that transfer better to unrelated tasks (segmentation, depth), not just ImageNet linear eval.

**Weaknesses:**

1. Central assumption relies on a rough linearization. The core trick (perturb in the left nullspace of the next block so the next block's output is unchanged) is only feasible because they linearize the non-linear transformer block using the FFN weights and then pick the smallest singular vectors. But real blocks have attention, non-linearities, layernorm, residual couplings, and cross-token interactions. The paper cites an appendix showing FFN is the dominant source of norm amplification, but in the main text this is still an approximation, not a guarantee. A reviewer could fairly ask: how often does the adapter actually stay in the true (nonlinear) null directions during training? The ablation in Table 4a helps, but it’s still empirical.

2. Evaluation scope is narrow. All main results are ViT-Large to ViT-Tiny, canonical ViT, every-second-layer alignment. That is a single, quite specific distillation path. There is no evidence it works for: (i) teacher and student of different families (e.g., DINOv2 to DeiT-Small), (ii) CNN to ViT, or (iii) same-size or near-size distillation where artifacts might behave differently. This makes the generality claim ("artifact-aware KD for VFMs") weaker than it sounds.

3. Choice of baselines. They compare to FitNet and ViTKD, but they do not compare to the most natural artifact-aware teacher-side fixes like register tokens (Darcet et al., 2024), or to SINDER / DINOv3-style cleaned teachers as sources for distillation--both of which are already in their related work. If the claim is "we are the principled way to keep info while suppressing artifacts", then we need to see: what if you just gave the student a cleaner teacher in the first place? Right now that experiment is missing.

4. Table 1 is confusing / formatting issues. The version here has lines that appear interleaved or misaligned (e.g., "SiNGER 70.59 8.16 21.76 3.03 ..."), which makes it hard to confirm the exact gains on each benchmark. For a distillation paper, exact numbers matter. This hurts Presentation.

5. Cost and practicality not fully discussed. Attaching adapters to multiple teacher layers and running SVD-style initialization on (linearized) FFNs is not free. The paper doesn’t quantify the extra compute / memory during distillation vs. plain feature KD. This is important for practitioners.

6. Long-tail result potentially goes in the wrong direction. The iNat2019 result is weaker and is brushed off as “long-tail is hard,” but that’s precisely a setting where better supervision and less artifact noise should help the student pick up minority classes. A short analysis or per-class breakdown would strengthen the story.

7. Causality of the improvement is still a bit entangled. Three things change at once: teacher refinement, outlier loss, and Gram matching. We do see an ablation on Linfo, but we don’t see, for example, "plain LoRA adapter without nullspace init" vs "nullspace init" vs "only percentile shrinkage" on all tasks, just on a subset. It’s still plausible that a well-tuned teacher-side LoRA denoiser would get close.

**Questions:**

See weakness.

---

> ### Author Response · Authors · 2025-11-20
> **Official Author Response to Reviewer YFDi (1/5)**
>
> # 1.  Validity of FFN-based Linearization and Empirical Nullspace Verification
>
> > W1. Central assumption relies on a rough linearization. The core trick (perturb in the left nullspace of the next block so the next block's output is unchanged) is only feasible because they linearize the non-linear transformer block using the FFN weights and then pick the smallest singular vectors. But real blocks have attention, non-linearities, layernorm, residual couplings, and cross-token interactions. The paper cites an appendix showing FFN is the dominant source of norm amplification, but in the main text this is still an approximation, not a guarantee. A reviewer could fairly ask: how often does the adapter actually stay in the true (nonlinear) null directions during training? The ablation in **Table 4a** helps, but it’s still empirical.
>
>  We appreciate the reviewer’s valid concern regarding the complexity of real Transformer blocks. We fully acknowledge that our FFN-based linearization is a simplification of the intricate architecture involving Self-Attention, LayerNorm, and residual couplings. However, as noted in our **Appendix A**, our empirical analysis identified the FFN sub-layer as the principal source of norm inflation, which motivated this design choice.
>
>  To address your concern about whether the adapter stays in the "true" null directions, we provide a two-fold verification:
>
> a. **Internal Consistency (Alignment with Computed Nullspace)** : First, we verified in Table 4a of the manuscript that our adapter successfully maintains high alignment with the computed nullspace throughout training. Specifically, the alignment metric $E_{safe}$ reaches 0.83 with our method, compared to <0.27 for random initialization. This confirms that our optimization strategy works as intended: the adapter stays close to the initialization provided by our linearization.
>
> b. **External Validity (Verification against Non-linear Block)** : We agree that internal consistency is only meaningful if the computed nullspace is a valid proxy for the true non-linear nullspace. Since analytically defining the "ground truth" nullspace of the full non-linear block is intractable, we designed a **Nullspace Verification Experiment** to empirically validate our proxy.
>
> c. **Nullspace Verification Experiment** : We compared the safeness of the nullspace computed by our method (SiNGER) against a baseline derived from the full Jacobian of the block.
>
> 1. **Setup**: We used a pre-trained ViT-Large block $W$ and a subset of 5,000 ImageNet images $D$.
> 2. **Methods**:
>
> * SiNGER: Nullspace $N'$ computed from our FFN-linearization $W'$ (without using dataset $D$).
> * Jacobian: Nullspace $N''$ computed from the Jacobian $W''$ estimated on the same dataset $D$.
>
> 3. **Metric**: We perturbed the input $D$ along the computed null directions to generate $D' = D + \Delta_{N'}$ (SiNGER) and $D'' = D + \Delta_{N''}$ (Jacobian). We then measured the deviation between the original output $F = W(D)$ and the perturbed outputs $F' = W(D')$ and $F'' = W(D'')$. A "true" nullspace should yield minimal change between the original and perturbed outputs.
>
> | Metric | FFN-based $W'$ |Jacobian-based $W''$|
> |:-|:-|:-|
> | $L_2$ diff ($\downarrow$) | 0.169 | 0.191 |
> | Cosine sim ($\uparrow$) | 0.9787 | 0.9654 |
> | CKA ($\uparrow$) | 0.9975 | 0.9947 |
>
>  The results demonstrate that perturbations guided by our FFN-linearized nullspace result in negligible changes to the actual non-linear block's output. Notably, our method induces even less output deviation than perturbations based on the full block Jacobian estimated on the same data. While we acknowledge that defining a global nullspace for general non-linear functions remains theoretically intractable, this experiment empirically suggests that our FFN-centric linearization serves as a highly robust proxy. It confirms that, within the scope of our distillation objective, the computed subspace effectively preserves the teacher's functional behavior, thereby validating the practical utility of the approximation challenged in the review.

---

> ### Author Response · Authors · 2025-11-20
> **Official Author Response to Reviewer YFDi (2/5)**
>
> # 2. Experiment scope expansion.
>
> > W2. Evaluation scope is narrow. All main results are ViT-Large to ViT-Tiny, canonical ViT, every-second-layer alignment. That is a single, quite specific distillation path. There is no evidence it works for: (i) teacher and student of different families (e.g., DINOv2 to DeiT-Ⅲ-Small), (ii) CNN to ViT, or (iii) same-size or near-size distillation where artifacts might behave differently. This makes the generality claim ("artifact-aware KD for VFMs") weaker than it sounds.
>
> We thank the reviewer’s observation regarding the limited evaluation scope. Following this suggestion, we conducted three additional experiments to broaden the teacher–student configurations, covering near-sized distillation, different transformer family (DeiT-Ⅲ), and cross-family transfers (DINOv2 to DeiT-Ⅲ). Although some experiments were run on a reduced subset of ImageNet-1K due to computational constraints, all results consistently support the generality of SiNGER.
>
> | Teacher | Student | Distillation | IN-1K val $(\uparrow)$ | ADE-20K $(\uparrow)$ | NYUd-v2 $(\downarrow)$ |
> |:-|:-|:-|:-:|:-:|:-:|
> | ViT-Large | ViT-Small | FitNet | 72.50 | 25.15 | 0.9903 |
> | ViT-Large | ViT-Small | SiNGER | 79.13 | 30.06 | 0.9026 |
>
> Even when the teacher and student capacities are close, SiNGER outperforms FitNet across classification, segmentation, and depth estimation. This suggests that SiNGER is not dependent on large teacher-student gaps.
>
> | Teacher | Student | Distillation | IN-1K val $(\uparrow)$ | ADE-20K $(\uparrow)$ | NYUd-v2 $(\downarrow)$ |
> |:-|:-|:-|:-:|:-:|:-:|
> | DeiT-Ⅲ-Large | DeiT-Ⅲ-Base | FitNet | 60.00 | 26.79 | 1.1625 |
> | DeiT-Ⅲ-Large | DeiT-Ⅲ-Base | ViTKD | 66.54 | 19.58 | 1.2525 |
> | DeiT-Ⅲ-Large | DeiT-Ⅲ-Base | SiNGER | 79.37 | 29.47 | 1.1514 |
> | DeiT-Ⅲ-Small | DeiT-Ⅲ-Tiny | FitNet | 51.41 | 16.06 | 1.2920 |
> | DeiT-Ⅲ-Small | DeiT-Ⅲ-Tiny | ViTKD | 35.56 | 8.71 | 1.4487 |
> | DeiT-Ⅲ-Small | DeiT-Ⅲ-Tiny | SiNGER | 63.50 | 20.67 | 0.9827 |
>
> To strengthen the claim that SiNGER is stronger than ViTKD, we distilled the DeiT-Ⅲ in which was experimented in ViTKD paper. As a result, we found that ViTKD works with DeiT-Ⅲ family, winning FitNet in partial cases. SiNGER consistently surpassed both FitNet and ViTKD.
>
> | Teacher | Student | Distillation | IN-1K val $(\uparrow)$ | ADE-20K $(\uparrow)$ | NYUd-v2 $(\downarrow)$ |
> |:-|:-|:-|:-:|:-:|:-:|
> | DINOv2-Large | DeiT-Ⅲ-Tiny | FitNet | 2.90 | 2.86 | 1.2328 |
> | DINOv2-Large | DeiT-Ⅲ-Tiny | ViTKD | 2.84 | 2.54 | 1.2856 |
> | DINOv2-Large | DeiT-Ⅲ-Tiny | SiNGER | 3.40 | 2.91 | 1.2222 |
>
> Following the reviewer's insightful comment, we conducted cross-family distillation of DINOv2 to DeiT-Ⅲ. We would like to highlight the performance gain over FitNet and ViTKD because the distilled model is suboptimal due to the limited scale of the training dataset. SiNGER achieved approximately 17.2%, 1.75%, and 0.86% performance gain in ImageNet-1K validation, ADE-20K, and NYUd-v2 benchmarks, respectively.
>
> Across all expanded settings - near-sized ViTs, different ViT families, and cross-family transfer - SiNGER consistently outperforms both FitNet and ViTKD, demonstrating that the method is effective beyond the originally presented ViT-L to ViT-T scenario. These results strengthen our claim that SiNGER is broadly applicable as an artifact-aware distillation framework for Vision Foundation Models. We will update both the main paper and the appendix to include the additional experiments and to better emphasize the applicability of SiNGER beyond the original setting.

---

> ### Author Response · Authors · 2025-11-20
> **Official Author Response to Reviewer YFDi (3/5)**
>
> # 3. Choice of baselines.
>
> > W3. Choice of baselines. They compare to FitNet and ViTKD, but they do not compare to the most natural artifact-aware teacher-side fixes like register tokens (Darcet et al., 2024), or to SINDER / DINOv3-style cleaned teachers as sources for distillation--both of which are already in their related work. If the claim is "we are the principled way to keep info while suppressing artifacts", then we need to see: what if you just gave the student a cleaner teacher in the first place? Right now that experiment is missing.
>
> We appreciate the reviewer for pointing this out. To address this concern, we conducted additional experiments using clean-teacher baselines, including (1) DINOv2 with register tokens (DINOv2-reg) and (2) DINOv3. Due to computational constraints, these experiments were performed on a small subset of ImageNet-1K, but all methods were trained under identical settings.
>
> | Teacher | Student | Distillation | IN-1K val $(\uparrow)$ | ADE-20K $(\uparrow)$ | NYUd-v2 $(\downarrow)$ |
> |:-|:-|:-|:-:|:-:|:-:|
> | DINOv2-reg-Large | ViT-Tiny | FitNet | 6.97 | 3.06 | 1.3711 |
> | DINOv2-reg-Large | ViT-Tiny | ViTKD | 1.81 | 1.57 | 1.3499 |
> | DINOv2-reg-Large | ViT-Tiny | SiNGER | 6.10 | 2.78 | 1.4058 |
> | DINOv3-Large | ViT-Tiny | FitNet | 1.84 | 1.16 | 1.3868 |
> | DINOv3-Large | ViT-Tiny | ViTKD | 0.23 | 0.42 | 1.3157 |
> | DINOv3-Large | ViT-Tiny | SiNGER | 1.19 | 1.06 | 1.4550 |
>
> We would like to clarify that SiNGER is designed under the assumption that high-norm artifacts are present in the teacher’s representations.
> Our artifact-suppression loss reduces the excessive norm of such tokens; however, when the teacher already has minimal artifacts (e.g., DINOv2-reg, DINOv3), this suppression may partially attenuate useful vectors, leading to a small drop in absolute performance.
> That said, even in these unfavorable settings, SiNGER remains more stable than ViTKD, which relies on random masking and often behaves inconsistently when artifact structure is weak or absent. We view this as evidence that SiNGER provides a more principled and less brittle form of teacher-side refinement.
> We agree that an adaptive suppression mechanism, which modulates refinement strength depending on the artifact level of each token or layer, would make SiNGER more robust across both artifact-heavy and artifact-light teachers. We consider this a promising direction for future work.
>
> # 4. Readability of Table 1.
>
> > W4. Table 1 is confusing / formatting issues. The version here has lines that appear interleaved or misaligned (e.g., "SiNGER 70.59 8.16 21.76 3.03 ..."), which makes it hard to confirm the exact gains on each benchmark. For a distillation paper, exact numbers matter. This hurts Presentation.
>
> We appreciate the reviewer for highlighting the readability issues in Table 1. To address this, we will reorganize the table structure to make the comparisons clearer. Specifically, we will (1) move all gain values into a separate row to avoid misalignment, (2) include the full benchmark results (DS, FG) in the appendix due to space constraints, and (3) replace ambiguous model abbreviations with explicit names (e.g., ViT-L → ViT-T), with notation clarified in the caption. These changes will make the table easier to interpret while maintaining all relevant comparisons.
>
> | Model | Distillation | IN-val | ADE-20K | NYUd-v2 | iNat2019 | DS | FG |
> |:-|:-|:-:|:-:|:-:|:-:|:-:|:-:|
> | ViT-L | | 79.58 | 26.57 | 0.9157 | 71.42 | 54.20 | 82.27 |
> | ViT-T | | 58.03 | 14.20 | 1.1807 | 43.95 | 28.75 | 62.02 |
> | ViT-L $\rightarrow$ ViT-T | FitNet | 62.43 | 18.73 | 1.0093 | 40.02 | 32.32 | 62.48 |
> | | ViTKD | 5.07 | 11.92 | 1.1903 | 23.69 | 2.08 | 33.52 |
> | | SiNGER | 70.59 | 21.76 | 0.9406 | 41.11 | 38.87 | 64.61 |
> | | $\Delta$ (gain) | 8.16 | 3.03 | 0.0687 | -2.84 | 6.55 | 2.13 |
>
> These revisions will be reflected in the main paper and the appendix.

---

> ### Author Response · Authors · 2025-11-20
> **Official Author Response to Reviewer YFDi (4/5)**
>
> # 5.  Computational Cost and Practicality Analysis
>
> > W5. Cost and practicality not fully discussed. Attaching adapters to multiple teacher layers and running SVD-style initialization on (linearized) FFNs is not free. The paper doesn’t quantify the extra compute / memory during distillation vs. plain feature KD. This is important for practitioners.
>
> We thank the reviewer for raising this important practical point. We agree that for a distillation method to be widely adoptable, its computational and memory overhead must be justified. We have quantified the costs associated with SiNGER in our experimental setup (ViT-L Teacher $\to$ ViT-T Student, distilling at 2 layers) and summarize them as follows:
>
> a. **Initialization Cost** : The SVD-based initialization is a one-time process performed before training starts. Computing the nullspace for the linearized weights takes approximately 0.085 seconds per layer. For our setup with two distillation layers, the total initialization time is **under 0.2 seconds**, which is effectively negligible in the context of the entire training pipeline.
>
> b. **Memory Overhead** : **SiNGER** introduces additional parameters only via the lightweight LoRA-based adapters ($d=1024, r=16$). Total added parameters are $2 \text{ layers} \times 2 \text{ adapter(up/down)} \times (1024 \times 16) = 65,536$ parameters. Compared to the ViT-Tiny student (~5.5M parameters), this represents an increase of only ~1.2%. This fits easily within standard GPU memory budgets.
>
> c.  **Training Compute Overhead** :   We observed an approximate **10% increase** in training time per epoch compared to the FitNet baseline. We break down the source of this overhead using Big-O notation (where $n$: patches, $d$: hidden dim, $r$: rank, $\phi$ : adapter):
>
> 1. **Adapter Forward ($O(ndr)$)**: The operation $\hat{F} = F + (F\phi_{down})\phi_{up}$ requires $O(ndr)$ FLOPs. Since $r \ll d$, this cost is minimal compared to the attention mechanism $O(n^2d)$.
> 2. **Outlier Loss ($O(nd)$)**: Computing Outlier Loss involves simple norm calculations, which are computationally inexpensive.
> 3. **Information Loss (Primary Bottleneck)**: This objective requires passing the refined features $\hat{F}_l$ through the teacher's next block $W$ ($l+1$th).
>
> - **Extra Forward Pass**: This incurs the cost of a full ViT block, roughly $O(n^2d + nd^2)$. For a Large Teacher ($d=1024$), this is the dominant additional cost.
> - **Gram Matrix**: Calculating $G(\hat{F}_{l+1})$ adds $O(n^2d)$.
>
> While the current overhead is manageable, we acknowledge that the Information Preservation Loss constitutes the primary computational cost. For scenarios where training speed is important, we envision that this overhead could be mitigated through future optimizations.
> For instance, the computational cost of the Gram matrix calculation could likely be reduced by computing relations only on a random subset of patches rather than the full set. Furthermore, to avoid the extra forward pass entirely, one might explore directly penalizing the perturbation's alignment with the teacher's principal basis (computed at initialization). We believe these directions could effectively enforce the nullspace constraint with lower computational complexity and plan to explore their validation in future work.

---

> ### Author Response · Authors · 2025-11-20
> **Official Author Response to Reviewer YFDi (5/5)**
>
> # 6. Weakness of distillation in long-tail learning.
>
> > W6. Long-tail result potentially goes in the wrong direction. The iNat2019 result is weaker and is brushed off as “long-tail is hard,” but that’s precisely a setting where better supervision and less artifact noise should help the student pick up minority classes. A short analysis or per-class breakdown would strengthen the story.
>
> We appreciate the reviewer for raising this point. To better understand the long-tail behavior, we analyzed the entropy of the teacher’s logits across full dataset, major, and long-tail subsets of iNaturalist2019. A clear pattern emerges:
>
> | | Full (53,649 samples) | | Major (1,873 samples) | | Long-tail (1,819 samples) | |
> |:-|:-|:-|:-|:-|:-|:-|
> | Model | Entropy | Accuracy | Entropy | Accuracy | Entropy | Accuracy |
> | ViT-Large | 6.6336 | 71.41 | 6.6290 | 74.21 | 6.6526 | 61.24 |
> | ViT-Tiny | 6.6435 | 44.71 | 6.6421 | 48.10 | 6.6548 | 32.22 |
> | SiNGER | 6.6940 | 41.08 | 6.6933 | 48.37 | 6.6989 | 24.96 |
> | FitNet | 6.7097 | 39.98 | 6.7078 | 44.69 | 6.7135 | 26.50 |
>
> The number in the parenthesis is the number of samples.
> The teacher (ViT-Large) shows the largest entropy increase specifically in long-tail classes, indicating substantially lower confidence and higher prediction ambiguity in minority categories.
> Since distillation transfers the teacher’s class-level certainty, this heightened uncertainty is directly inherited by the student.
> While SiNGER suppresses spatial artifacts effectively, it does not modify the teacher’s class logits. As a result, the long-tail accuracy becomes limited not by artifact noise but by the teacher’s inherently uncertain supervision for rare classes. This explains why the student trained with SiNGER shows smaller gains—or even lower accuracy—than a Tiny model trained without distillation, despite improving overall performance.
> Importantly, SiNGER still produces lower long-tail entropy than FitNet (6.6989 vs. 6.7135), suggesting that our refinement mechanism yields more stable supervision even when the teacher’s uncertainty dominates.
> We will incorporate this clarification into the revised version.
>
> # 7. On the causality of the improvement.
>
> > W7. Causality of the improvement is still a bit entangled. Three things change at once: teacher refinement, outlier loss, and Gram matching. We do see an ablation on Linfo, but we don’t see, for example, "plain LoRA adapter without nullspace init" vs "nullspace init" vs "only percentile shrinkage" on all tasks, just on a subset. It’s still plausible that a well-tuned teacher-side LoRA denoiser would get close.
>
> A clearer separation of effects across refinement components is indeed valuable for understanding the source of gains. To make this explicit, we conducted an expanded ablation that isolates the SiNGER adapter itself, the role of nullspace initialization, and the contributions of the outlier suppression (percentile shrinkage) and information preservation (Gram-matching) losses. The results are summarized below.
>
> | Initialization | $\mathcal{L}_\text{outlier}$ | $\mathcal{L}_\text{info}$ | IN-1K val $(\uparrow)$ | ADE-20K $(\uparrow)$ | NYUd-v2 $(\downarrow)$ |
> |:-|:-:|:-:|:-:|:-:|:-:|
> | random | | | 10.83 | 4.35 | 1.3048 |
> | nullspace | | | 10.57 | 4.32 | 1.1943 |
> | nullspace | ✓ | | 11.04 | 4.77 | 1.1870 |
> | nullspace | | ✓ | 11.06 | 4.38 | 1.1870 |
> | nullspace | ✓ | ✓ | 11.57 | 4.99 | 1.1690 |
>
> The extended ablation illustrates how each component contributes to the overall refinement quality. Switching from random to nullspace-based initialization stabilizes the refinement direction and avoids injecting perturbations that conflict with the next block’s behavior. Adding the outlier-suppression loss provides the largest gain in both ImageNet-1K and ADE-20K, as it directly targets the dominant artifact tokens that bias distillation. Gram matching further improves information consistency. When all components are combined, the student achieves its strongest performance across tasks, indicating that SiNGER’s design works most effectively as an integrated framework rather than a single isolated mechanism.
> These results clarify the individual contribution of each design choice, and the full ablation will be included in the main paper.

---

### Author Response · Authors · 2025-11-27
**Notification of Revised Manuscript**

Dear Reviewers,

We sincerely thank you for the time and thoughtful feedback during the discussion phase.
We have uploaded a revised version of the manuscript reflecting your suggestions.
All substantial updates are highlighted in dark green.

# Summary of Key Updates
## Main Manuscript

- **Clarified Scope**: Clearly stated our scope of interest as visual transformers (§. 2, L111).
- **Diversified Baselines**: Included DeiT-III as a main baseline (§. 4.1, §. 4.2, Tab. 1).
- **Extended Ablation**: Added a full component ablation study (§. 4.5, Tab. 4).
- **Discussion**: Added discussion section on the nullspace approximation gap and the complexity of SiNGER (§. 5).

## Appendix

- **Architectural Specs**: Added expanded architectural details reflecting the broader baseline set (Appx. B).
- **Diverse Teacher-Student Pairs**: Added experiments with cleaner teachers, heterogeneous architectures, and a larger teacher model (Appx. C).
- **Long-Tail Learning**: Added theoretical interpretation on the performance in long-tail learning (Appx. J).

We hope these revisions address your main concerns and would greatly appreciate your re-evaluation of the updated manuscript.
We believe the paper has been significantly strengthened and welcome any further questions.

Sincerely,

The Authors

---

### Meta-Review · Area_Chair_ZETc · 2025-12-28

**Summary:**

The paper presents a method for improving Vision Transformer distillation by refining teacher features to suppress high-norm artifacts. The motivation is reasonable, the approach is clearly described, and the rebuttal’s Jacobian-based verification helps clarify the theoretical soundness of the core formulation.

However, many of the additional experiments and ablations rely on ImageNet-1K subset settings with low absolute performance, making the results difficult to interpret and limiting the strength of the empirical evidence. While Table 1 reports large gains over FitNet, the rebuttal analyzes component-wise contributions only in subset-based settings with low absolute performance and small performance gaps, making it difficult to establish clear causal links between the proposed components and the reported improvements. In addition, the applicability of the method to recent artifact-cleaned teachers (e.g., DINOv3) or non-ViT architectures remains limited. We encourage these limitations to be more carefully clarified and discussed in the camera-ready version.

Nevertheless, the reviewers’ scores are relatively high, and the problem and proposed method are well motivated. Overall, despite the above limitations, the contribution is considered meaningful, and based on the reviews and rebuttal, I recommend Accept.

**Reviewer Concerns:**

The rebuttal effectively addressed several presentation issues and provided useful additional evaluations, including new teacher–student settings, Jacobian-based verification, and a cost analysis.

The reviewers noted limitations in the general applicability of the proposed method (e.g., to CNNs or hybrid architectures), as it primarily targets ViT-specific artifacts. The rebuttal does not provide validation on non-ViT architectures and leaves such extensions to future work. In addition, it reports reduced effectiveness on recent models such as DINOv3 (cleaner teachers). Taken together, these observations suggest that the practical applicability of the proposed approach is currently limited to a narrower set of scenarios.

Ablation studies are essential to validate the motivation and effectiveness of the proposed method. While the rebuttal addresses reviewers’ concerns, several key ablations are conducted only on ImageNet-1K subsets, making their impact harder to assess. Given the large performance gap between SiNGER and FitNet reported in Table 1, more carefully designed and full-scale ablation experiments would be beneficial for substantiating the claimed causal contributions.

I encourage the authors to address these concerns and to more clearly discuss them as limitations in the camera-ready version.

**Reviewer Scores:**

This submission received initial scores of 6, 6, and 6 from the three reviewers. In response to the reviewers’ comments, the authors made substantial revisions and added many experiments, addressing several important concerns.

---

### Decision · Program_Chairs · 2026-01-26

Accept (Poster)